# Reinforcement Learning Teachers of Test Time Scaling

**Edoardo Cetin, Tianyu Zhao, Yujin Tang**
Sakana AI, Japan
{edo,tianyu,yujintang}@sakana.ai

## Abstract

Training reasoning language models (LMs) with reinforcement learning (RL) for one-hot correctness inherently relies on the LM being able to explore and solve its task with some chance at initialization. Furthermore, a key use case of reasoning LMs is to act as teachers for distilling new students and cold-starting future RL iterations rather than being deployed themselves. From these considerations, we introduce a new framework that avoids RL's exploration challenge by training a new class of Reinforcement-Learned Teachers (RLTs) focused on yielding the most effective downstream distillation. RLTs are prompted with both the question and solution to each problem, and tasked to simply "connect-the-dots" with detailed explanations tailored for their students. We train RLTs with dense rewards obtained by feeding each explanation to the student and testing its understanding of the problem's solution. In practice, the raw outputs of a 7B RLT provide higher final performance on competition and graduate-level tasks than existing distillation and cold-starting pipelines that collect and postprocess the reasoning traces of orders of magnitude larger LMs. Furthermore, RLTs maintain their effectiveness when training larger students and when applied zero-shot to out-of-distribution tasks, unlocking new levels of efficiency and re-usability for the RL reasoning framework.

 github.com/SakanaAI/RLT

## 1   Introduction

Exploration is one of the critical challenges in reinforcement learning (RL) and has been a core focus of its literature [1–3]. Sparse rewards cannot yield any learning signal unless the agent is already capable of solving the given task at initialization. With the rise of RL for open-ended reasoning (RL reasoning) inducing a new form of language model (LM) scaling [4–6] beyond prompt-engineering and search [7, 8], exploration has reemerged as a key challenge. A canonical motivation for RL is the potential to bootstrap from partial solutions, guided by an informative reward function, and learn entirely new tasks from scratch. However, the nature of one-hot correctness rewards used in the RL

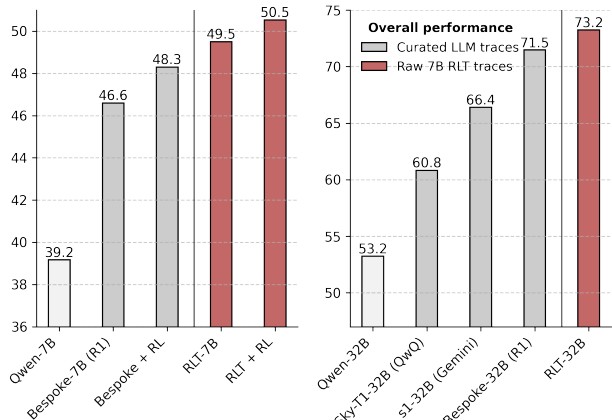

Figure 1: RLTs provide better student distillation and RL cold-starts than orders of magnitude larger LMs across competition and graduate-level tasks (AIME, MATH, GPQA). This holds when distilling students of the same size (Left) and also 32B students, much larger than the RLT itself (Right).

39th Conference on Neural Information Processing Systems (NeurIPS 2025).

reasoning framework fails to provide a dense form of guidance to assess relative progress, focusing instead on reinforcing correct responses in the initial model's pool of pass-at-k attempts – without true extrapolation beyond the LM's initial latent abilities [9]. As a result, mostly large, already-capable models have been shown to improve consistently beyond cheaper and simpler supervised optimization [4].

Due to this fundamental limitation, coupled with RL's training instability, distillation has emerged as another ubiquitous component of current reasoning paradigms. In this case, the test-time role of LMs trained with RL is to act as a *teacher* providing instructive reasoning traces for a *student* to solve new problems. This teacher-student paradigm is widely adopted both to train smaller, less-capable models [6, 10] and even to cold-start future RL iterations for better final convergence with the teacher's own initial checkpoint acting as the student [4, 11]. However, the problem-solving skills reinforced by correctness-based rewards have been shown not to be entirely aligned with the goal of downstream distillation [12]. To account for this mismatch, current pipelines significantly rely on heuristic-driven postprocessing of the teacher's outputs for effective student transfer [4, 6, 12, 13].

Based on these considerations, we propose a framework that avoids RL's exploration challenge with a new class of specialized **Reinforcement-Learned Teachers (RLTs)** trained specifically to yield effective downstream distillation. Our main intuition is simple: the ability of real-world teachers is not measured by whether they can come up on their own with complex theorems, proofs, or answers from scratch. Instead, what matters is their ability to make use of readily available solutions and devise instructive explanations for their students. Thus, we depart from the traditional RL reasoning framework, tasking a model to first **think** and then come up with a new **solution** for the first time. Instead, RLTs are tasked with the easier problem of providing an effective **explanation** with the problem's **solution** already given within their prompt. We train RLTs with dense and informative rewards obtained from the student's log probabilities. These rewards provide an intuitive measure of the student's understanding of each problem's ground-truth solution following the teacher's explanations, and the interpretability of the logical leaps in the explanations themselves.

By distilling students from the raw outputs of a lightweight RLT with 7B parameters, we demonstrate significantly higher performance than using existing pipelines relying on reasoning LMs with orders of magnitude more parameters (Figure 1). We show our framework provides superior benefits even when distilling the RLT's explanation to train larger 32B students and to cold-start traditional RL optimization. Furthermore, we showcase how RLTs can be transferred zero-shot to new domains and still produce effective distillation datasets that yield yet better final students than direct RL with access to the task's reward. Overall, these results highlight the potential of our new method for overcoming the large costs of RL by focusing on stronger, smaller, and highly reusable specialized teachers, while removing the current reliance on expensive and heuristic-driven distillation pipelines.

We share our code and pretrained checkpoints[1] to facilitate future research in RL reasoning and distillation. In summary, our main contributions are threefold:

- We introduce the RLT framework, tackling exploration with a simpler dense-reward that aligns the objective of RL training to providing effective downstream student distillation.

- We show how distilling the raw outputs of a 7B RLT directly outperforms training students with carefully postprocessed reasoning traces from orders of magnitude larger LMs.

- We demonstrate that RLTs also allow for better cold-starts for traditional RL, effective distillation to larger students, and even zero-shot transfer to new reasoning domains.

## 2 Inducing reasoning in language models

### 2.1 Reinforcement learning

The RL post-training recipe for inducing reasoning behavior was recently popularized by the DeepSeek R1 line of work [14, 15, 4]. By fine-tuning on a dataset of questions $D = \{q_1, \ldots, q_N\}$ with verifiable solutions $\{s_1, \ldots, s_N\}$, Guo et al. [4] show effective "reasoning" behavior emerges out of a 671B-parameter LM [16], significantly pushing its performance on challenging math and coding tasks. Their training is conducted with GRPO [15], an online RL algorithm that foregoes

---

[1] `https://github.com/SakanaAI/RLT`

Figure 2: Left: RL format asking an LM to think and solve hard problems from scratch. Right: RLT format asking an LM to produce instructive step-by-step explanations given access to the solutions.

the use of a critic model with a simple Monte-Carlo value estimate. GRPO prompts the LM $\pi_\theta$ to produce a set of $G >> 1$ "grouped" outputs $o_1, ...o_G$ for each sampled question $q \in D$, optimizing:

$$J(\theta) = \mathbb{E}_{q \sim D, \{o\}_1^G \sim \pi_\theta(\cdot|q)} \left[ \frac{1}{G} \sum_{i=1}^{G} \left( A_i - \beta \, \mathbb{D}_{\mathrm{KL}}(\pi_\theta \, \| \, \pi_{\mathrm{ref}}) \right) \right]. \tag{1}$$

Here, the "advantages" $A_i$ are obtained by normalizing each output's reward $r_i$ within each group:

$$A_i = \frac{r_i - \mathrm{mean}(\{r_1, \ldots, r_G\})}{\mathrm{std}(\{r_1, \ldots, r_G\})}. \tag{2}$$

A key component of their design is a system prompt that asks the LM to decompose each generated output $o_i$ into two separate formatted sections separated by `<think>` and `<solution>` tags, denoted $t_{o_i}$ and $s_{o_i}$. This structure is forced by assigning rewards $r_i = -1$ to unformatted completions, $r_i = -0.5$ to wrong but formatted completions, and $r_i = 1$ only to correct and formatted completions. Training with this strategy, Guo et al. [4] show the LM's completion length gradually grows with reflection, verification, and self-correction steps emerging, mirroring human chain-of-thoughts.

## 2.2 Supervised distillation

Supervised distillation is another critical step used to train recent reasoning models to complement RL's shortcomings. For any online RL objective like Equation 1 to avoid collapse, the model must already possess a non-trivial chance of producing correct responses with non-zero gradients at initialization. This defining property makes the RL objective much less applicable than cross-entropy objectives that always include the correct response's information in the model's gradients. As a consequence of this dichotomy, distilling the reasoning traces of large RL-trained LMs with supervised learning is not only cheaper but has also been shown to be notably more effective than performing RL itself for inducing reasoning in smaller, less-capable models [4, 6, 17, 12, 10]. Furthermore, RL appears prone to instabilities and output degradation, especially during extended training sessions. Due to this second limitation, DeepSeek R1 and several other models [4, 11] perform RL training over multiple iterations. This is done by using the RL-trained models at the end of each intermediate iteration only to, once again, collect distillation datasets used for "cold-starting" their original initial checkpoint and obtain a stronger initialization point for the next RL iteration.

Constructing a dataset of distillation prompts $D_{SD} = \{d_1, \ldots, d_N\}$ involves using the RL-trained LM $\pi_\theta$ with its reasoning system prompt to answer a corpus of verifiable questions, which can be chosen with several heuristics [6, 17, 12, 10]. The LM's output reasoning traces for each question $o \sim \pi_\theta(\cdot|q)$ are then filtered by comparing them with the ground-truth solutions to ensure their correctness. Commonly, these reasoning traces are also post-processed via additional "manual" steps of refinements, such as asking other closed-sourced LMs to remove grammatical issues and refactor the reasoning steps into a nicer, consistent format. In fact, Li et al. [12] even argues that the structure and format of the thinking data is a critical component to make weaker models actually understand and learn how to reason from distillation, potentially even more important than correctness itself.

# 3 Reinforcement learning teachers

## 3.1 The implications of training teacher models as students

We distinguish two separate training and inference "roles" that can be performed by LMs in the modern reasoning framework. As detailed in Section 2, after RL training, LMs $\pi_\theta$ are often not deployed themselves but rather used to obtain reasoning distillation datasets for fine-tuning weaker models and cold-starting future RL iterations. Thus, these models can be effectively seen as **teachers**, providing **explanations** for future **student** models $\pi_s$ to learn from.

This teacher-student paradigm highlights a potential mismatch between the objective used for RL training and the teacher's test-time role. In traditional settings, teachers are trained with sparse correctness rewards to improve their ability to solve hard problems from scratch. This objective not only precludes the applicability of RL training for tasks beyond the base model's original capabilities, due to its inherent exploration challenge, but is also not aligned with the teacher's actual end goal: producing reasoning traces from which students $\pi_s$ can learn the necessary skills to derive correct solutions themselves. Based on these considerations, we propose a different training framework for RL reasoning models to be deployed as teachers that avoids RL's exploration challenge and breaks this objective mismatch. Our framework comprises a much easier task formulation, a dense reward objective, and a carefully designed training recipe, allowing us to learn a new class of specialized Reinforcement Learned Teachers (RLTs).

## 3.2 Aligning the task of teacher models

In the traditional RL paradigm, the solution $s_i$ to each problem is never explicitly provided to the model and is only employed for checking the correctness of the corresponding solutions within the LM's completions $s_{o_i}$. Precluding direct access or information about the solutions aligns training with the test-time objective of solving entirely new test problems from scratch, but is precisely what makes exploration challenging, as the model receives no gradients until its first successful attempt. Our key observation, however, is that the test-time "teaching" objective of producing effective distillation datasets $D_{\mathrm{SD}}$ for questions with known solutions, can be greatly facilitated by explicitly providing access to such solutions – as is the case for real-world teachers, who can rely on access to readily available solutions and, thus, focus entirely on how instructive their explanations are for students.

To this end, as illustrated in Figure 2, RLTs are prompted with a new formatting style, providing both the question and solution to each problem as inputs, and are tasked to produce instructive step-by-step explanations, connecting the dots between the two. We design our prompts to allow direct reuse of the teacher's outputs for student distillation while keeping the task natural, appending the solution tokens $s_i$ and tags to the RLT's system prompt and input question before generating each completion. At test-time,

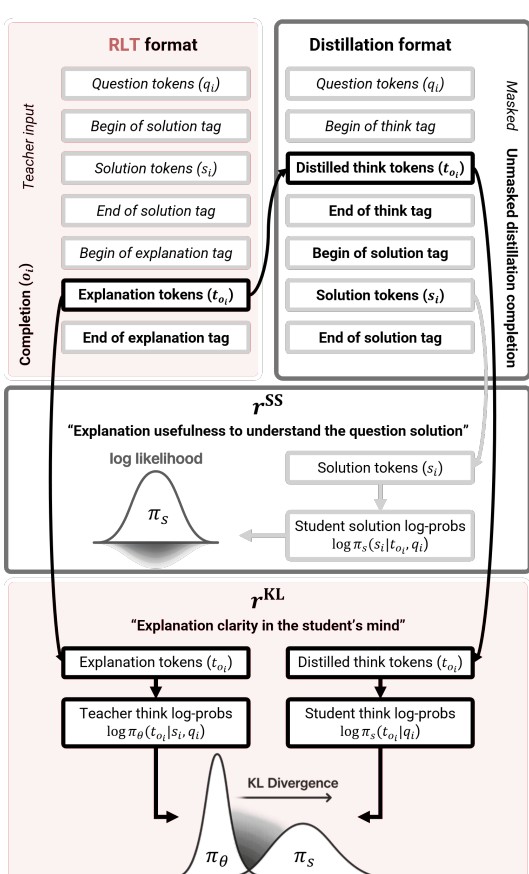

Figure 3: The tokens from the RLT's explanations are copied into the student format to measure its understanding with our reward terms.

constructing the corresponding question completions for the student distillation datasets $d_i \in D_{\mathrm{SD}}$ is then as simple as extracting the think tokens from the teacher's outputs by replacing surrounding `explanation` with `think` tags and appending back the solutions $s_i$.

## 3.3 Evaluating the quality of explanations

The reward function to train RLTs is made of two terms to incentivize explanations $o_i$ that lead the student $\pi_s$ to recover correct solutions $s_i$ and are also logical continuations from questions alone under the student's perspective. In particular, following the procedure from the previous subsection, for each completion $o_i$ from the teacher $\pi_\theta$, we extract the think tokens $t_{o_i}$ and format the corresponding student distillation prompt $d_i$ by prepending the question $q_i$ and appending the ground-truth solution $s_i$. As illustrated in Figure 3, each distillation prompt is then fed as input to the student model to obtain a set of per-token log probabilities, which are processed into our two reward terms as follows:

i. $r_i^{\text{SS}}$: quantifying the student $\pi_s$ understanding of the solutions $s_i$ given the question $q_i$ and think tokens $t_{o_i}$ in context. This first reward term is computed with the student's log probabilities over the solution tokens. We reduce this vector to a scalar by applying both average and minimum operations over the different per-token values:

$$r^{\text{SS}}(o_i, s_i, q_i) = \text{avg}\left\{\log \pi_s^{s_i}\right\} + \alpha \min \left\{\log \pi_s^{s_i}\right\}, \quad \text{where} \quad \pi_s^{s_i} = \pi_s(s_i \mid t_{o_i}.q_i). \quad (3)$$

ii. $r_i^{\text{KL}}$: quantifying whether the think tokens $t_{o_i}$ themselves are interpretable logical continuations from the student's perspective as compared with the teacher's. This second reward term is computed with the KL divergence over the same think tokens between the teacher's distribution (under the RLT's format with both $q_i$ and $s_i$ in context) and the student's (with only the question $q_i$ in context). Similarly to $r_i^{\text{SS}}$, we reduce this vector to a scalar by applying both average and maximum operations over the different per-token values:

$$r^{\text{KL}}(o_i, s_i, q_i) = \text{avg}\left\{\mathbb{D}_{\text{KL}}\left(\pi_\theta^{t_{o_i}} \| \pi_s^{t_{o_i}}\right)\right\} + \alpha \max\left\{\mathbb{D}_{\text{KL}}\left(\pi_\theta^{t_{o_i}} \| \pi_s^{t_{o_i}}\right)\right\}, \quad (4)$$

$$\text{where} \quad \pi_s^{t_{o_i}} = \pi_s(t_{o_i} \mid q_i), \quad \pi_\theta^{t_{o_i}} = \pi_\theta(t_{o_i} \mid s_i, q_i).$$

Finally, the RLT rewards are obtained by combining these two terms with a weighting coefficient $\lambda$:

$$r_i^{\text{RLT}} = r^{\text{SS}}(o_i, s_i, q_i) - \lambda r^{\text{KL}}(o_i, s_i, q_i) \quad (5)$$

Each term in our reward function serves a precise purpose. First, optimizing $r^{\text{SS}}$ will produce explanations containing think tokens $t_{o_i}$ that maximize the student's likelihood of reaching the correct solution $s_i$. However, this term alone does not differentiate between explanations that guide the student step-by-step and those that increase the solution's likelihood without a logical path that can be learned from. An extreme instance of the latter would be an explanation simply repeating the solution tokens to increase likelihood, failing to provide the student with general examples of reasoning methods that can be applied when approaching new problems. Thus, introducing $r^{\text{KL}}$ fills precisely this gap, aligning the teacher's distribution toward the student's such that each think token in the output explanations cannot have too low probability when formatted in the distillation prompt $d_i$ with only the question $q_i$ and the previous think tokens in context.

Intuitively, introducing this term regularizes for each step in the logical path traced by the teacher's explanation to still make sense in the "student's mind" given only its prior understanding and the question itself. We note that if we instead compared two distributions conditioned on the question alone, the KL would vanish, and $r^{KL}$ would fail to penalize the RLT steps that the teacher could not have generated without the solution in context. Additionally, combining the average with min/max reduction terms ensures the rewards do not forego any individual token, regardless of the solution length or the number of think tokens in the teacher's explanations. For instance, their omission could bias $r^{\text{SS}}$ based on the length of the solutions or lead the teacher to prefer long explanations only to reduce the influence on $r^{\text{KL}}$ of hard but necessary individual logical steps. For further discussion, we refer to Appendix D, where we empirically analyze and validate each design choice in our reward function in terms of final performance (Table 14) and concrete qualitative differences of the resulting explanations (Figures 8 through 12).

## 3.4 Pulling everything together: the RLT training paradigm

The RLT framework can be used with any RL algorithm (e.g., [18, 19]) with minimal modifications to the LM's conditioning and reward, as described in the above subsections. For all our main

Table 1: RLTs and prior distillation pipelines across model (7B and 32B) and data size (1K and 17K).

| Model | Data size | AIME 2024 | MATH 500 | GPQA Diamond | Overall |
|---|---|---|---|---|---|
| QwQ-32B | N.A. | 50.00 | 90.60 | 54.50 | 65.03 |
| DeepSeek-R1 | 800K+ | **79.80** | **97.30** | **71.50** | **82.87** |
| Qwen2.5-7B-Instruct | N.A. | 10.00 | 74.20 | 33.30 | 39.17 |
| Bespoke-7B-1K | 1K | 13.30 | 80.00 | 33.80 | 42.37 |
| RLT-7B-1K (Ours) | 1K | **20.00** | **80.40** | **41.90** | **47.43** |
| Bespoke-7B | 17K | 20.00 | 82.00 | 37.80 | 46.60 |
| RLT-7B (Ours) | 17K | **23.30** | **82.80** | **42.40** | **49.50** |
| Qwen2.5-32B-Instruct | N.A. | 26.70 | 84.00 | 49.00 | 53.23 |
| s1-32B | 1K | 50.00 | 92.60 | 56.60 | 66.40 |
| s1-32B + budget forcing | 1K | 56.70 | 93.00 | 59.60 | 69.77 |
| Bespoke-32B-1K | 1K | 46.70 | 92.60 | 57.50 | 65.60 |
| RLT-32B-1K (Ours) | 1K | **60.00** | **94.00** | **60.10** | **71.37** |
| Sky-T1-32B | 17K | 43.30 | 82.40 | 56.80 | 60.83 |
| Bespoke-32B | 17K | 63.30 | 93.00 | 58.10 | 71.47 |
| RLT-32B (Ours) | 17K | **66.70** | **93.40** | **59.60** | **73.23** |

experiments, we employ the simple GRPO recipe detailed in Section 2, resulting in the following training objective:

$$J^{\text{RLT}}(\theta) = \mathbb{E}_{q,s\sim D,\,\{o\}_1^G\sim\pi_\theta(\cdot|s,q)}\left[\frac{1}{G}\sum_{i=1}^{G}\left(A_i^{\text{RLT}} - \beta\,\mathbb{D}_{\text{KL}}(\pi_\theta\,\|\,\pi_{\text{ref}})\right)\right],\qquad(6)$$

where $A_i^{\text{RLT}}$ is computed with the normalization strategy defined in Equation 2 using the RLT reward function from Equation 5. Unlike for correctness-based rewards, our learning signal is inherently dense, providing informative rankings to the RLT's output even before achieving any task expertise. This fundamental difference greatly facilitates our optimization, akin to how heuristically shaped rewards enabled RL agents to learn entirely new behaviors for videogames and robotics tasks [20, 21].

## 4 Experiments

### 4.1 Training, distillation, and evaluation

We train RLTs on the set of questions and solutions selected by Li et al. [12] based on their level of challenge. This dataset comprises less than 17K math and coding problems originally used for distilling filtered and post-processed reasoning traces collected from QwQ [22] and DeepSeek R1 [4]. In contrast, the RLTs we consider are orders-of-magnitude smaller models, all trained starting from the Qwen2.5-7B-Instruct LM [23]. We precede our RL phase with a short supervised fine-tuning phase to familiarize RLTs with their new system prompt and input format using the open reasoning dataset released by Labs [13]. During RL, we compute the reward for the RLT explanations using another small Qwen-7B model as the student. We train our main models for 125 steps, less than a single epoch, with a batch size of $1024$, a constant learning rate of $1\times10^{-6}$, and a group size of $64$. We note that we were also able to train RLTs with a smaller batch size of $256$ and more steps for faster preliminary experimentation with only slightly inferior results.

We collect our distillation dataset with the learned RLTs using the same full set of 17K question-solution pairs from training. With the new reasoning traces, we then proceed to fine-tune our students either on this full data or a randomly sampled 1K subset, equating the distillation budget and following the same recipes as our baselines [6, 12]. Unlike previous RL distillation pipelines, we do not apply extra postprocessing refinements to improve the quality of the RLT's reasoning traces, directly using our model's raw outputs for student fine-tuning. We note this is in contrast to prior distillation pipelines that make use of the ground-truth answers to verify the correctness of their reasoning traces and rely on multi-generations, filtering, and post-processing stages to improve data quality [24, 13]. We refer to Appendices A and B for further details regarding our training and distillation phases with complete lists of hyperparameters.

Table 2: RLTs and prior distillation pipelines for cold-starting traditional RL.

| Model | Data size | AIME 2024 | MATH 500 | GPQA Diamond | Overall |
|---|---|---|---|---|---|
| Qwen2.5-7B-Instruct | N.A. | 10.00 | 74.20 | 33.30 | 39.17 |
| Bespoke-7B | 17K | 20.00 | 82.00 | 37.80 | 46.60 |
| RLT-7B (Ours) | 17K | **23.30** | **82.80** | **42.40** | **49.50** |
| RL no cold-start | N.A. | 13.30 | 74.20 | 34.80 | 40.77 |
| RL cold-start (raw) + RL | 17K | 10.00 | 71.00 | 34.80 | 38.60 |
| RL cold-start (GPT) + RL | 17K | 16.70 | 78.20 | 36.90 | 43.93 |
| Bespoke-7B + RL | 17K | 16.70 | 82.80 | **45.40** | 48.30 |
| RLT-7B + RL (Ours) | 17K | **26.70** | **84.00** | 40.90 | **50.53** |

Following prior work [6, 13, 12], our main evaluation considers three popular and challenging tasks from the literature: AIME24 [25], the set of problems used for the American Invitational Mathematics Examination. MATH 500 [26], the set of problems selected by [27] from the canonical competition math benchmark. GPQA Diamond [28], the set of diamond difficulty problems on natural science topics from the Graduate-level Google-proof Q&A benchmark. We report the completion accuracy of each of our students using Lighteval [29]. When available, we use baseline results reported in prior work, which we found close to our early reproduction attempts. In Appendix C, we extend the experiments in this section by evaluating our models on additional tasks and analyzing the impact of teacher size, the base RL algorithm, and the student used during training.

## 4.2 Test-time reasoning across teachers and students

Our main experiments focus on grounding the effectiveness of RLTs to obtain instructive reasoning traces beyond traditional distillation pipelines. As described in Section 4.1, to construct the student distillation dataset, we use the same starting question-solution pairs as our recent state-of-the-art baselines [12, 13], with each sample only differing in terms of its reasoning trace. While RLTs could be inexpensively applied to provide explanations of larger corpora, this consistency serves to remove potential confounding factors, other than the quality of the reasoning traces, biasing our experiments and comparisons. For the same reason, we do not retune any hyperparameters for the distillation phase, training students following the same procedure as our baselines based on data size [12, 6].

We compare the RLTs' explanations with prior approaches, evaluating students fine-tuned on both our full 17K distillation samples and its 1K subset. Our recent baselines all follow a similar recipe of distilling datasets obtained by generating reasoning traces with expensive reasoning models or API calls and postprocessing them with closed-source LMs: s1 [6] using traces from Gemini Flash Thinking [30], Sky-T1 [12] using traces from QwQ [22], and Bespoke [13] using traces from DeepSeek R1 [4]. Since the Bespoke baseline obtained state-of-the-art results with our same question-solution corpus, we extend its evaluation with new results distilling its processed R1 traces only for our same 1K questions subset, equating its number of datapoints with the other s1 baseline.

As shown in Table 1, the raw output explanations of our small 7B parameter RLT outperform all the considered data-distillation pipelines involving teachers with orders of magnitude more parameters and additional ad-hoc postprocessing steps. We also find that the effectiveness of the RLT traces stays consistent across different data sizes. In contrast, the R1 traces from the Bespoke pipeline appear significantly less effective when subsampled. Furthermore, even when distilling a Qwen-32B student, much larger than our 7B teacher, our RLT still outperforms all prior methods for both data sizes with considerable margins. We believe this result, in particular, shows how our framework could allow overcoming the current prohibitive costs of RL reasoning: shifting the burden of expensive RL procedures to small teachers, unable to effectively solve problems from scratch but highly specialized in the simpler task of producing effective explanations for large, more powerful students.

## 4.3 RLTs to cold-start RL

Our next set of experiments focuses on evaluating the effectiveness of RLTs in providing cold-start data for traditional RL. For this new RL phase, we use our same GRPO implementation with the standard student format and correctness-based rewards described in Section 2. As compared to the RLT framework, we find that using a larger batch size of 1024 is significantly more beneficial to

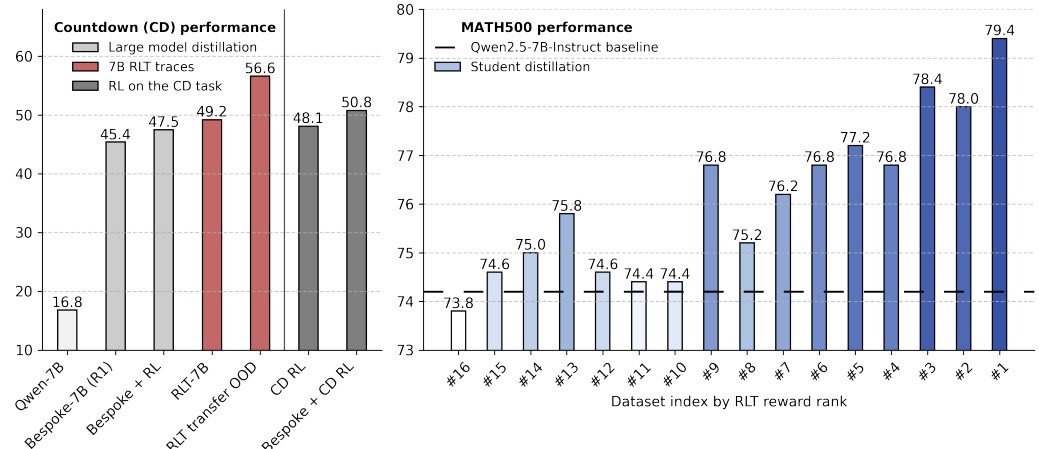

Figure 4: Left: Out-of-distribution performance transferring RLTs to produce new distillation data as compared to students trained on the Li et al. [12] corpus and direct RL on the countdown task. Right: Performance after training on different distillation datasets ranked by the RLT reward.

better cope with the increased variance and reward sparsity of traditional RL. We train for a full epoch on the recent RL dataset from Li et al. [31] collected by analyzing and selecting an effective subset of the competition math data based on the correlation of individual samples with overall performance improvement.

We compare performing this new RL phase on the Qwen-7B model cold-started from the reasoning traces of our 7B RLT and the postprocessed R1 traces from the Bespoke pipeline, our strongest distillation baseline. Moreover, we also compare a 7B parameters baseline teacher trained with traditional RL as done in prior work [11, 4]: effectively performing RL twice on the Qwen model and collecting a dataset at the end of the first iteration to cold-start the second. To construct the cold-starting dataset for this last baseline, we consider either taking the model's raw output traces, as done with RLTs, or postprocessing them with additional refinements using GPT4.1-mini [32] and following a very similar strategy to the other R1 and QwQ traditional distillation pipelines [12, 24].

As shown in Table 2, the reasoning traces from our 7B parameter RLT again display superior cold-starting effectiveness compared to all of our baselines. The performance gap is exceedingly noticeable with the cold-starting approaches that are also using a 7B teacher trained with traditional RL. In fact, only after improving the format and structure of the traces from these RL-trained teachers with GPT postprocessing, we were able to observe any improvements from the original Qwen-7B results. While our RLT was itself trained from the same 7B model, it again demonstrates superior cold-starting even when compared to postprocessed R1 pipelines. Overall, we find these results to be compelling evidence indicating that RLTs have the potential to unlock new key avenues to democratize the RL reasoning framework beyond the current reliance on prohibitively large and closed-source LMs.

### 4.4 Out-of-domain zero-shot transfer

Unlike problem-solving from scratch, we posit that providing effective explanations to given solutions is a much less task-specific skill. Thus, in this subsection, we evaluate how well RLTs can be applied to construct datasets and distill new specialized students in out-of-distribution domains, without any expensive RL retraining. In particular, we focus on the canonical countdown task [33], asking student models to combine a set of numbers to equal a given target using basic arithmetic operations. We train and test our models on distinct datasets of 16K and 1K automatically-generated question and solution pairs. We compare zero-shot transferring our RLT with transferring the RLT-7B student and the Bespoke-7B baselines from the previous subsections. To ground our results, we also consider performing RL on the countdown task itself (CD RL), training both from the Qwen-7B model and the cold-started Bespoke-7B baseline with the same setup described in Section 4.3.

As shown in the left bar plot of Figure 4, applying RLT distillation zero-shot remarkably achieves even higher performance than direct RL on the countdown task. Interestingly, direct RL appears to provide only marginally better scores than using students distilled from our original set of reasoning questions that do not include any examples of countdown problems (50.8 vs. 49.2). Furthermore, we

| **R1 postprocessed reasoning** | **RLT explanation** |
|---|---|
| `<|begin_of_thought|>` 

 […] 

 *Just to check, if I use **a calculator**…* 

 […] 

 *Another angle (**no pun intended**): using the derivative…* 

 […] 

 *Another way to check: maybe using complex numbers or other identities, **but I think the method is solid**… the final answer is…* 
 `<|end_of_thought|>` | `<|begin_of_explanation|>` 

 […] 

 *First, the unit circle definition…* 

 […] 

 *Alternatively, using calculus. The derivative of…* 

 […] 

 *Alternatively, use complex numbers. Let me try that. Express the two angles as complex numbers on the unit circle…* 
 `<|end_of_explanation|>` |

Figure 5: Examples comparing the contents from the post-processed R1 traces [13] that were particularly improved by the corresponding RLT explanations as measured by our reward function

find there is stark overlap of over 98.5% in the final sets of solved problems between direct RL and the RL-free Bespoke-7B baseline. We find these results in line with prior analysis [12, 9], providing further evidence that the exploration challenge of traditional RL might make most of its benefits come from steering the base model's distribution toward long-context generation. In contrast, by simplifying the task and foregoing sparse rewards, our RLT appears much more effective – providing countdown-specific traces that allow students to learn new knowledge and solve unseen questions, yielding higher improvements than direct RL even without any teacher training in this new domain.

## 4.5 Explanation reward analysis

To analyze the design of the RLT reward function, we start by examining the relationship between the traces' rewards and the effectiveness of student distillation. In particular, we use our RLT's checkpoint right before RL training to generate 16 completions for each question-solution pair in our data. We then score all completions with our reward and divide them into groups based on their relative rank for each prompt. Thus, we obtain 16 datasets with different reasoning traces for each question, which we use to train 16 new 7B students from Qwen. As illustrated in the right bar plot of Figure 4, ordering student performance by the respective dataset rank shows a clear correlation between the two, with a Pearson coefficient over 0.89, validating the efficacy of the RLT objective. Additionally, the highest ranked traces of our 7B teacher before any RL remarkably already yield 90% the performance gains of our baseline R1 distillation pipeline [13], showing how even small models already possess latent teaching skills unlocked by our new reward and simplified task formulation.

We also inspect qualitative examples chosen by selecting samples where the reward of the RLT explanations is particularly improved from the baseline R1 distillation pipeline. As shown in Figure 5, we find that the R1 traces with low rewards often try to rely on external tools, such as calculators, and employ language patterns likely idiosyncratic to the training data of the DeepSeek-V3 LM, as sentences with brief humorous comments [16]. Instead, the corresponding RLT explanations appear much more grounded and even manage to add new alternative verification steps not considered by R1 to check the final solution. In Appendix D, we provide additional examples showcasing further qualitative differences of our framework with R1 traces and also specific failure cases from training RLTs without proper balance between each reward component, such as repetitions and overly-long explanations.

## 5 Related work

Inspired by the unprecedented abilities of the OpenAI o1 model [34], there has been a resurgence of RL approaches aimed at inducing a new kind of open-ended reasoning to scale test-time compute. The work from Guo et al. [4] was another milestone in this new domain, providing a first openly detailed example of what is possible with large models and RL. Other follow-ups considered smaller LMs and ways to decrease optimization costs via approaches such as explicit task breakdown [11], exploration strategies [5], new RL objectives [35], and cold-start data scale [10]. However, it is still an open question if RL on small models can go beyond cheaper supervised alternatives [4] and induce new skills beyond the pretraining corpus [9]. In contrast to this work, RLTs break the traditional framework of maximizing one-hot accuracy with verifiable rewards – turning the task on its head by feeding the model the correct solution as input and avoiding RL's inherent exploration challenge.

A large part of the recent test-time scaling literature considering smaller LMs has focused on inducing reasoning with "teacher-student" supervised distillation [36], a widely validated technique in traditional LM development [23, 37]. This approach's popularity to induce LM reasoning dates earlier than the RL paradigm, with older methods harnessing verifiers and prompting for self-improvement [38, 39]. Part of this earlier exploration showed preliminary signs of how teachers could be considerably steered to provide better student data [40, 41] and how improvements in teaching also led to improvements in question-answering domains [42]. Lately, by following a common structure of generation, filtering correct responses, and postprocessing them, modern RL-based distillation has seen significant advances mostly driven by more capable teachers [13, 12] and carefully curating targeted datasets [6, 17]. However, the effectiveness of current distillation pipelines was shown to be closely tied to the properties of the student itself [43, 10], and their ability to induce actual generalization remains unclear [44]. Unlike these traditional distillation pipelines, the RLT framework does not rely on verifiers for filtering, directly optimizes the teacher for downstream distillation, and does not require any post-processing, allowing direct transfer of reasoning capabilities to arbitrary tasks and even larger student models.

## 6    Discussion and extensions

This work introduced a new class of Reinforcement-Learned Teachers trained with a simpler dense-reward task that inputs both each problem's question and solution, and optimizes the LM to provide instructive reasoning traces for distillation as outputs. Empirically, students trained or cold-started from the raw outputs of a 7B RLT obtain higher performance than prior distillation pipelines using orders-of-magnitude larger LMs. Furthermore, RLTs maintain their effectiveness also when providing reasoning traces for out-of-distribution tasks beyond their training corpus, and even for distilling much larger students than the teacher itself. These remarkable properties can have significant implications for the scalability of developing reasoning models, due to the disproportionate cost of RL compared to all other post-training stages: while distilling a 32B student on fixed traces took less than a day on a single H100 compute node, training this larger model using RL on the same questions would have taken months with the same hardware. Nonetheless, our work has only begun to study the design space of our new framework, with many exciting directions yet to be explored. One example is training RLTs and their students in tandem, allowing the teacher's explanations to adapt to the student's learning dynamics live. Pushing this further, the same model could even take both roles, iterating RL with our task formulation, providing access to each problem's solution, to obtain instructive step-by-step reasoning traces, and self-distillation [38] to revise its own explanation and learn how to solve questions from scratch – unifying the open-endedness of RL with the stability of supervised optimization.

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

Table 3: Hyperparameter listing for the RLT training optimization and reward.

| Hyperparameter name | Value |
|---|---|
| **RLT training** | |
| Fine-tuned model | Qwen2.5-7B-instruct [23] |
| Number of training steps | 125 |
| Batch size | 1024 |
| Learning rate | $1 \times 10^{-6}$ |
| Learning rate decay | Constant |
| Final learning rate | $1 \times 10^{-6}$ |
| Weight decay | 0 |
| Optimizer | AdamW [45] |
| Adam beta1 | 0.9 |
| Adam beta2 | 0.999 |
| Adam epsilon | 1e-8 |
| Warmup steps | 0 |
| Maximum gradient norm | 1.0 |
| Maximum generation context size | 16384 |
| Generation temperature | 0.7 |
| Generation top-$p$ | 1.0 |
| Generation top-$k$ | No |
| Generation min-$p$ | 0.0 |
| Generation repetition penalty | 1.0 |
| GRPO group size | 64 |
| GRPO $\beta$ | 0.04 |
| Reference model sync. steps | 32 |
| Reference model sync. mixup | 0.9 |
| Dtype | bfloat16 |
| Gradient checkpointing | true |
| **RLT reward** | |
| Student model | Qwen2.5-7B-instruct [23] |
| $\lambda$ | 3 |
| $\alpha$ | 0.01 |
| Format penalty | -1 |

# A  Implementation details

## A.1  RLT training phase and reward

Our experiments are conducted on a single compute node comprising 8 Nvidia Hopper H100 GPUs, 1.8 TB of memory, and 208 Intel Xeon Platinum 8481C CPUs. Due to the efficiency of training with our small 7B models, we note this setup is significantly less resource-intensive than prior RL and even SFT work, often relying on multi-node settings [6, 12]. As described in Section 4, for its new RL phase, we train our Reinforcement Learned Teachers on the set of questions and solutions selected by Li et al. [12][2] which is available under an Apache 2.0 License, comprising less than 17K math and coding problems. All our new teachers are trained from a Qwen2.5-7B-Instruct LM [23] also available under an Apache 2.0 License, together with the other models from the Qwen family. Before RL, we shortly fine-tune on pre-collected samples using example traces from Labs [13] formatted using our new system prompt and tags. This short, inexpensive phase is conducted with the same distillation hyperparameters used for the 1K data subset following Muennighoff et al. [6], detailed in Appendix B, but with double the number of epochs, and serves to quickly familiarize our teacher

---

[2]https://huggingface.co/datasets/bespokelabs/Bespoke-Stratos-17k

Table 4: Cost to train and generate datasets with RLT models as opposed to DeepSeek R1 [4].

| Model/GPU-hours | Training (GPU model) | Data generation (GPU model) |
|---|---|---|
| DeepSeek R1 | >688000 (H800) | >1067 (H100) |
| 7B RLT teacher | 280.4 (H100) | 6.7 (H100) |

with the new RLT input format. We also use the output checkpoint at the end of this phase for our correlation analysis experiment conducted in Section 4.5. For the same reason and also to limit the requirements for reproducibility, our lightweight Qwen2.5-7B-instruct [23] student model used to compute the RLT reward function is initialized with the checkpoint provided by Labs [13][3] available under an Apache 2.0 License, which is already familiar with their system prompt without needing further training. However, we found that the RLT reward is robust to the specific LM choice for the student, yielding numerically close values when using our own Qwen student finetuned on only 1K samples.

We use simple coefficients to regulate the terms in the RLT rewards, with $\lambda = 3$ to scale $r^{\mathrm{KL}}$ and $\alpha = 0.01$ to scale both the min term in $r^{\mathrm{SS}}$ and the max term in $r^{\mathrm{KL}}$. Our choice was based on making each individual component of the RLT rewards have approximately the same expected magnitude over the model's initial output completions. As we observed the overall rankings to be quite robust after testing small changes to these initial choices, we did not find extensive sweeps necessary. We also add a -1 penalty to any completion that does not use the explanation tags or that exceeds our maximum generation context length to limit training time and disincentivize overly long and expensive reasoning traces. For our training runs, we make use of a custom GRPO implementation with the specifications from Shao et al. [15], extending the TRL library [46] with faster distributed VLLM generation [47]. Our RL phase is short, comprising only 125 steps, less than a single epoch, with a batch size of $1024$, an AdamW optimizer [45] with a constant learning rate of $1 \times 10^{-6}$, and a group size of $64$. We synchronize the reference model every 32 steps as popularized by [48] with a mixup ratio of 0.9. For the post-cold-start RL phase employed in Section 4.3, we use our same GRPO implementation with the standard student format and correctness-based rewards described in Section 2. The only difference is that we train for one epoch in total on the very small LIMR [31] dataset, containing less than 1K samples. We provide a full list of hyperparameters to ensure reproducibility in Table 3. All main libraries used in our implementation are again available under an Apache 2.0 Licence.

## A.2 Traditional reasoning and RLT formats

As detailed in Section 3.2 and illustrated in Figure 2, RLTs are prompted with a new formatting style that provides both the question and solution to each problem as inputs. Then, their system prompt instructs to output instructive, detailed step-by-step explanations, connecting the dots between the two. In contrast, traditional formats used for reasoning datasets employed for RL and distillation forego any information about each problem's solution and task the model to solve each problem from scratch. We provide a specific comparison between the two, contrasting the system prompts used for traditional RL and distillation from Li et al. [12] in Figure A.2, and our own new RLT input format in Figure A.2. As shown, our prompting format's design strives to make minimal changes to the prompts used for traditional reasoning frameworks, replacing surrounding `explanation` with `think` tags, and simply appending the solution tokens after the user's provided input question, to allow our teachers to make full use of this information before generating each completion. This design also practically avoids the need to re-prompt our teacher multiple times for each question and manually check the answers to filter out incorrect solutions.

## A.3 Cost comparison

In Table 4, we report the GPU cost of training and collecting distillation data with our RLT. We compared these times with an estimated cost of traditional RL post-training and data collection using the DeepSeek R1 model. To compute this estimate, we used official figures for the GPUs used to train this DeepSeek model (2,048 NVIDIA H800 GPUs [16]) and online communication from the DeepSeek team for the training time. We calculated a lower bound of the GPU hours for

---

[3]`https://huggingface.co/bespokelabs/Bespoke-Stratos-7B`

**System prompt**

```
<|im_start|>system
Your role as an assistant involves thoroughly exploring questions
↪   through a systematic long thinking process before providing the
↪   final precise and accurate solutions. This requires engaging in a
↪   comprehensive cycle of analysis, summarizing, exploration,
↪   reassessment, reflection, backtracing, and iteration to develop
↪   well-considered thinking process. Please structure your response
↪   into two main sections: Thought and Solution. In the Thought section,
↪   detail your reasoning process using the specified format:
↪   <|begin_of_thought|> {thought with steps separated with '\n\n'}
↪   <|end_of_thought|> Each step should include detailed considerations
↪   such as analisying questions, summarizing relevant findings,
↪   brainstorming new ideas, verifying the accuracy of the current steps,
↪   refining any errors, and revisiting previous steps. In the Solution
↪   section, based on various attempts, explorations, and reflections
↪   from the Thought section, systematically present the final solution
↪   that you deem correct. The solution should remain a logical,
↪   accurate, concise expression style and detail necessary step needed
↪   to reach the conclusion, formatted as follows: <|begin_of_solution|>
↪   {final formatted, precise, and clear solution} <|end_of_solution|>
↪   Now, try to solve the following question through the above
↪   guidelines:<|im_end|>
```

**Generation prefix**

```
<|im_start|>user
Return your final response within \boxed{}. Positive integers $a$ and
↪   $b$ are such that the graphs of $y=ax+5$ and $y=3x+b$ intersect the
↪   $x$-axis at the same point. What is the sum of all possible
↪   $x$-coordinates of these points of intersection?
$\textbf{(A)}\ {-20}\qquad\textbf{(B)}\ {-18}\qquad\textbf{(C)}\
↪   {-15}\qquad\textbf{(D)}\ {-12}\qquad\textbf{(E)}\ {-8}$<|im_end|>
<|im_start|>assistant
<|begin_of_thought|>
```

Figure 6: Reasoning input format employed in traditional RL and student distillation, using an example question and the system prompt from Li et al. [12], providing the model instructions first to present a step-by-step rationale and then the problem's solution, deriving it from scratch.

generating the distillation dataset by multiplying the minimum amount of GPU resources required to host DeepSeek R1 with half precision and the "roof inference speed" for generating each completion in our dataset, without taking into account slowdowns from the growing context size. However, we note that both these estimates should be considered large approximations as we did not even consider time/resources for multiple generations, filtering, and post-processing stages conducted in our baselines [12, 13].

## B  Student distillation

### B.1  Reasoning traces generation

As detailed in Sections 3 and 4, we collect reasoning traces for each sample in the student distillation dataset by feeding our RLTs both the problem's question and its solution as input. Moreover, for our

Figure 7: RLT Input format using an example question from Li et al. [12], providing the model instructions to produce a step-by-step explanation given the solution to each problem as input.

cold-starting experiments in Section 4.3, we also collect reasoning traces by feeding the RL-trained Qwen models each question and postprocessing each output with GPT4.1 [32] similarly to Li et al. [12]. We provide the hyperparameters used to collect these traces across all our datasets and settings in Table 5. In particular, we use standard generation hyperparameters for reasoning Qwen-based models [12], that are aligned with the online generation parameters during our new RL phases, as detailed in Appendix A. The only main difference is that we allow for a longer maximum context size to avoid collecting incomplete traces for downstream distillation, which is possible thanks to the fact that at test time we are not subject to the same training-time memory constraints dictated by backpropagating through long traces.

Table 5: Hyperparameter listing for the RLT generation pipeline.

| Hyperparameter name | Value |
|---|---|
| **Distillation data generation setting** | |
| Maximum generation context size | 32764 |
| Generation temperature | 0.7 |
| Generation top-$p$ | 1.0 |
| Generation top-$k$ | No |
| Generation min-$p$ | 0.0 |
| Generation repetition penalty | 1.0 |
| Generation dtype | bfloat16 |

Table 6: Hyperparameter listing student distillation phases on the datasets from the set of questions from Li et al. [12], and the countdown dataset.

| Hyperparameter name | Full fine-tuning | 1K subset fine-tuning | Countdown fine-tuning |
|---|---|---|---|
| **Student training** | | | |
| Distilled model | | Qwen2.5-7B/32B-instruct [23] | |
| Number of training samples | **16710** | **1000** | **16000** |
| Number of epochs | **3.0** | **5.0** | **3.0** |
| Batch size | **96** | **16** | **96** |
| Learning rate | $1 \times 10^{-5}$ | $1 \times 10^{-5}$ | $1 \times 10^{-5}$ |
| Learning rate decay | Cosine | Cosine | Cosine |
| Final learning rate | $1 \times 10^{-6}$ | $1 \times 10^{-6}$ | $1 \times 10^{-6}$ |
| Weight decay | **0** | $1 \times 10^{-4}$ | **0** |
| Optimizer | AdamW [45] | AdamW [45] | AdamW [45] |
| Adam beta1 | 0.9 | 0.9 | 0.9 |
| Adam beta2 | **0.999** | **0.95** | **0.999** |
| Adam epsilon | $1 \times 10^{-8}$ | $1 \times 10^{-8}$ | $1 \times 10^{-8}$ |
| Warmup ratio | **0.1** | **0.05** | **0.1** |
| Maximum gradient norm | 1.0 | 1.0 | 1.0 |
| Dtype | bfloat16 | bfloat16 | bfloat16 |
| Gradient checkpointing | true | true | true |

The RLT distillation datasets used for training all of our 7B students were collected by generating a single completion for each question-solution pair, directly placing it in the student format. After preliminary experiments, we also found our 32B students can be particularly sensitive to cropped reasoning traces that exceed the 16384 maximum context length specified in the SFT hyperparameters from Li et al. [12], which they purposefully limited for computational efficiency. Thus, to avoid this mismatch potentially affecting our results, we simply collected up to 16 reasoning traces for each question-solution pair and selected anyone below 16384, otherwise resorting to $r^{SS}$ for selection.

## B.2 Student distillation specifications

Our main experiments from Section 4 use the RLT traces collected either from the full set of training questions from Li et al. [12] or its randomly selected 1K data subset. For the student distillation phases of these experiments, we re-use the same hyperparameters from Li et al. [12] and Muennighoff et al. [6], for the full data and 1K subset, respectively. The purpose of not re-tuning based on our new data is to attempt to isolate the reasoning traces used for student distillation as the only degree of variation in our comparison between RLTs with traditional reasoning distillation pipelines.

Table 7: Hyperparameter listing for the traditional RL on the Li et al. [31] dataset.

| Hyperparameter name | LIMR data traditional RL | Countdown data traditional RL |
|---|---|---|
| **Traditional RL student training** | | |
| Fine-tuned model | Qwen2.5-7B-instruct [23]/Bespokes-7B [13] | |
| Number of training samples | **1389** | **16000** |
| Number of epochs | 1.0 | 1.0 |
| Number of training steps | **86** | **250** |
| Batch size | **1024** | **256** |
| Learning rate | $1 \times 10^{-6}$ | $1 \times 10^{-6}$ |
| Learning rate decay | Constant | Constant |
| Final learning rate | $1 \times 10^{-6}$ | $1 \times 10^{-6}$ |
| Weight decay | 0 | 0 |
| Optimizer | AdamW [45] | AdamW [45] |
| Adam beta1 | 0.9 | 0.9 |
| Adam beta2 | 0.999 | 0.999 |
| Adam epsilon | $1 \times 10^{-8}$ | $1 \times 10^{-8}$ |
| Warmup steps | 0 | 0 |
| Maximum gradient norm | 1.0 | 1.0 |
| Maximum generation context size | 16384 | 16384 |
| Generation temperature | 0.7 | 0.7 |
| Generation top-$p$ | 1.0 | 1.0 |
| Generation top-$k$ | No | No |
| Generation min-$p$ | 0.0 | 0.0 |
| Generation repetition penalty | 1.0 | 1.0 |
| GRPO group size | 64 | 64 |
| GRPO $\beta$ | 0.04 | 0.04 |
| Reference model sync. steps | 32 | 32 |
| Reference model sync. mixup | 0.9 | 0.9 |
| Dtype | bfloat16 | bfloat16 |
| Gradient checkpointing | true | true |

Furthermore, in our experiments in Section 4.4, we also compare transferring students learned on our original set of datapoints and direct RL with transferring the RLTs themselves zero-shot to the countdown task [33]. For these experiment, we use a set of 16K automatically-generated countdown question and solution pairs with 3 or 4 numbers and, after obtaining the corresponding RLT traces, distill our students using the hyperparameters from Li et al. [12] once again, which we found to work well in practice without further tuning, given the two similar dataset sizes.

We provide a full list of the hyperparameters used for all our distillation experiments in each setting in Table 6, where we highlight the key differences across the three. As detailed, the considered approaches mostly differ in terms of the number of epochs, the batch size, and the optimizer parameters. These differences reflect the total number of samples and relative variance in each data batch. In line with the findings from the relative prior works [6, 12, 24], we note that training on these small datasets was very inexpensive and could be completed within hours. We note that the learning rate of our reinforcement learning phases matches the final learning rate during our SFT optimization, a simple choice which we found to work well in practice.

## B.3 Student RL details

For our cold-starting experiments in Section 4.3, we also implement and perform a phase of traditional RL training optimizing the model with correctness-based rewards under the "student's perspective."

Table 8: Student evaluation hyperparameter listing for the RLT generation pipeline, matching the hyperparameters from Li et al. [12].

| Hyperparameter name | Value |
|---|---|
| **Student evaluation** | |
| Maximum generation context size | 32764 |
| Generation temperature | 0.7 |
| Generation top-$p$ | 1.0 |
| Generation top-$k$ | No |
| Generation min-$p$ | 0.0 |
| Generation repetition penalty | 1.0 |
| Generation dtype | bfloat16 |

This phase is done atop a Qwen2.5-7B-Instruct, the cold-started Bespoke-7B, and the RLT-7B models using our same GRPO implementation on the open-source LIMR dataset [31]. We re-use most of the hyperparameters from the RLT training phase with a batch size of 1024, something particularly important to cope with the increased variance and reward sparsity of traditional RL. Additionally, we also conduct traditional RL as a baseline for our out-of-distribution transfer experiments in Section 4.4 that directly optimizes correctness on the countdown task starting again from the Qwen2.5-7B-Instruct model, and the cold-started Bespoke-7B models. For this particular task, we found using a larger batch size to be not strictly necessary, as for RLT training, and obtained better results optimizing the model for more steps (250 total) with a batch size of 256. We provide a full list of the distillation hyperparameters used for our RL phases on these tasks in Table 7, where we highlight the key differences between these two settings.

### B.4 Student evaluation

As described in Section 4.1, our main evaluation consists of three graduate and competition-level tasks on math and natural science domains. In particular, these include AIME24 [25], the set of problems used for the American Invitational Mathematics Examination; MATH 500 [26], the set of problems selected by [27] from the canonical competition math benchmark; and GPQA Diamond [28], the set of diamond difficulty problems on natural science topics from the Graduate-level Google-proof Q&A benchmark. Additionally, in Appendix C, we also extend our set of experiments to include additional challenging coding and multilingual domains. In particular, we consider LiveCodeBench [49], a contamination-free set of coding challenge problems continuously collected from several prominent online hosting platforms; and OlympiadBench [50], a set of olympiad-level bilingual problems in English and Chinese from past math and physics competitions.

We evaluate on all the above benchmarks using Lighteval [29], a library available under the MIT license. In all our results, we report the completion accuracy of each of our students for a single generated completion, as also reported in prior work [6, 12, 13]. Furthermore, we also ensure consistency by re-using the task implementation code, including the system prompt, provided by our baselines [24], which is available under an Apache 2.0 License. For the same reason, we do not modify any of the existing evaluation generation hyperparameters from the suggested settings used in their evaluation, which we report in Table 8.

## C Additional experiments

### C.1 Coding and multilingual reasoning

We extend our main set of experiments from Section 4, focusing on graduate and competition-level tasks on math and natural science domains, comparing the effectiveness of the reasoning traces from our 7B parameter RLT with traditional distillation pipelines. We consider challenging coding and multilingual domains, which are less aligned with the training and distillation set of questions employed. In particular, these new benchmarks include LiveCodeBench (LCB) [49], a contamination-

Table 9: RLTs and prior distillation pipelines across model (7B and 32B) evaluated on coding and multilingual reasoning benchmarks. Overall is computed by taking the average between LCB-Average and OlympiadBench scores.

| Model | Data size | LCB-Average | LCB-Hard | OlympiadBench | Overall |
|---|---|---|---|---|---|
| Qwen2.5-7B-Instruct | N.A. | 31.88 | **3.30** | 35.90 | 33.89 |
| Bespoke-7B | 17K | **36.10** | 1.60 | 43.30 | 39.70 |
| RLT-7B | 17K | 34.63 | **3.30** | **46.10** | **40.37** |
| Qwen2.5-32B-Instruct | N.A. | 48.94 | 9.80 | 46.70 | 47.82 |
| Sky-T1-32B | 17K | 57.94 | 17.90 | 57.30 | 57.62 |
| Bespoke-32B | 17K | 71.06 | 26.20 | 60.30 | 65.68 |
| RLT-32B | 17K | **71.24** | **32.50** | **64.00** | **67.62** |

free set of coding challenge problems continuously collected from several prominent online hosting platforms across three difficulty categories; and OlympiadBench [50], a set of olympiad-level bilingual problems in English and Chinese from past math and physics competitions.

In Table 9, we provide a comparison of reported results from our baselines [12, 13] and our fine-tuned student models using our 7B RLT traces. We also re-collected the performance on OlympiadBench of the Qwen2.5-7B-Instruct model and the Bespoke-7B baseline fine-tuned from the postprocessed R1 traces, as they were omitted in the reference results from prior work. For LiveCodeBench, we report both the performance on the "hard" difficulty set of problems and the average weighted performance. This was obtained by weighting the performances of the models in the "easy," "medium," and "hard" sets of LiveCodeBench problems, by the relative number of problems in each.

We find the performance of RLT distilled students on these new tasks to be consistent with the performance from our Section 4 experiments. In particular, the overall performance exceeds the performance of the baseline distillation pipelines using orders of magnitude larger models across student sizes. Furthermore, the RLT performance is also best across the individual settings, with the sole exception of LCB-average only for the 7B model, where it comes as a close second. However, as shown by the experiments in Section 4.4, by equating the pool of initial questions from which to perform distillation, we believe these experiments could still be underplaying the true potential enabled by the zero-shot transferability of RLTs. In particular, transferring RLTs, rather than their students, to construct reasoning traces to include more coding and Chinese-written problems could allow downstream distillation to develop further domain-specific expertise and reasoning, without the need to run expensive pipelines requiring large and closed-source models.

## C.2 The generality of RLT across RL algorithms

Table 10: Student performance from RLTs trained with GRPO [15], and RLOO [19].

| Evaluated model | Data size | AIME 2024 | MATH 500 | GPQA Diamond | Overall |
|---|---|---|---|---|---|
| Qwen2.5-7B-Instruct | N/A | 10.00 | 74.20 | 33.30 | 39.20 |
| Bespoke-7B | 17K | 20.00 | 82.00 | 37.80 | 46.60 |
| RLT-7B | 17K | **23.30** | 82.80 | 42.40 | **49.50** |
| RLT-7B (RLOO) | 17K | 20.00 | **83.60** | **42.90** | 48.80 |

To validate the generality of the RLT framework, we extended our implementation by training our 7B RLT model using the RLOO [19] algorithm instead of GRPO. We provide this comparison in Table 10 above. In line with recent empirical findings about the similar empirical effectiveness of the different reasoning RL algorithms [51], the performance of our RLT trained with RLOO appears very close to our results obtained using GRPO in the main text. While there exists a small gap, we believe this is mostly due to the fact that we did not re-adjust any of the main hyperparameters from our GRPO implementation (e.g., batch size, number of generations per question). These results provide further concrete evidence confirming that the RLT paradigm's performance is not tied to any specific RL algorithm.

Table 11: Student performance from RLTs with 3B and 7B parameters.

| Evaluated model | Data size | AIME 2024 | MATH 500 | GPQA Diamond | Overall |
|---|---|---|---|---|---|
| Qwen2.5-7B-Instruct | N/A | 10.00 | 74.20 | 33.30 | 39.20 |
| RLT-7B | 17K | **23.30** | **82.80** | **42.40** | **49.50** |
| RLT-7B (3B teacher) | 17K | 20.00 | 80.60 | 38.90 | 46.50 |
| Qwen2.5-32B-Instruct | N/A | 26.70 | 84.00 | 49.00 | 53.20 |
| RLT-32B | 17K | **66.70** | **93.40** | **59.60** | **73.20** |
| RLT-32B (3B teacher) | 17K | 46.70 | 91.40 | 53.50 | 63.90 |

## C.3 The effects of teacher scale

To validate the effects of the RLT teacher's scale, we extended our implementation by training an RLT model with 3B parameters. The only change to our hyperparameters was to double the number of supervised finetuning iterations, but kept the same number of RL training steps. As shown in Table 11, we expectedly find that larger teachers yield better explanations and downstream results. However, we find that most of the performance gap between our 3B and 7B teachers occurs when distilling the 32B student LM, where the mismatch between teacher and student capabilities tends to the extreme. However, even in this case where the student is over 10x larger than the teacher, our 3B RLT is still able to provide considerable improvements to the initial Qwen 32B student. The reason that we did not observe performance degradation appears to be due to the role of the $r^{KL}$ reward term plays during the optimization. Even in cases where the initial 3B teacher is unable to provide logical explanations to a specific question and answer, optimizing $r^{KL}$ will naturally guide the teacher's output distribution (with both question and answer in context) toward converging to what was most likely from the student's distribution (with only the question in context).

## C.4 Stronger students make stronger teachers

Table 12: Student performance using RLTs trained with a combination of 7B and 32B students, and following a 7B student updated midway through training.

| Evaluated model | Data size | AIME 2024 | MATH 500 | GPQA Diamond | Overall |
|---|---|---|---|---|---|
| Qwen2.5-7B-Instruct | N/A | 10.00 | 74.20 | 33.30 | 39.20 |
| Bespoke-7B | 17K | 20.00 | 82.00 | 37.80 | 46.60 |
| RLT-7B | 17K | 23.30 | 82.80 | **42.40** | 49.50 |
| RLT-7B (7B + 32B) | 17K | 23.30 | 83.80 | 41.90 | 49.70 |
| RLT-7B (2-stage) | 17K | **26.70** | **84.00** | 41.40 | **50.70** |

We extended our implementation to analyze how scaling up student capabilities to compute the terms in the RLT reward affects the training of our new models. First, we considered ensembling both the 7B and the 32B students, averaging their probabilities, thus also introducing a degree of diversity into the RLT reward. Second, we considered conducting training in two stages, each optimizing both teacher and student for half the total number of optimization steps. In particular, we paused the teacher RL optimization midway, distilled a student with this preliminary RLT's explanations, and resumed our teacher optimization for the remaining steps with this updated student to compute the RLT reward. This second extension has the added benefit of better aligning the teacher rewards to the student's learning dynamics, and can be considered an initial step toward online optimization of both models, which we believe to be an interesting future direction, as outlined in Section 6. As shown in Table 12 above, both these extensions provide noticeable improvements to the RLT distillation performance. Overall, these results, together with the ones showing the effects of teacher scale from Table 11, further highlight the complementarity of our new framework to harness future models and LLM advances across different beneficial axes.

Table 13: The effects of data size from training with RLT-generated data from different numbers of questions from Li et al. [12] (1K and 17K) and its NuminaMath split [52].

| Evaluated model | Data size | AIME 2024 | MATH 500 | GPQA Diamond | Overall |
|---|---|---|---|---|---|
| Qwen2.5-7B-Instruct | N/A | 10.00 | 74.20 | 33.30 | 39.20 |
| Bespoke-7B-1K | 1K | 13.30 | 80.00 | 33.80 | 42.40 |
| Bespoke-7B | 17K | 20.00 | 82.00 | 37.80 | 46.60 |
| Bespoke-7B (1K NuminaMath) | 1K | 16.70 | 79.00 | 33.30 | 43.00 |
| Bespoke-7B (10K NuminaMath) | 10K | 16.70 | 79.60 | 40.90 | 45.70 |
| RLT-7B-1K | 1K | 20.00 | 80.40 | 41.90 | 47.40 |
| RLT-7B | 17K | **23.30** | **82.80** | 42.40 | **49.50** |
| RLT-7B (1K NuminaMath) | 1K | 16.70 | 80.40 | 40.40 | 45.80 |
| RLT-7B (10K NuminaMath) | 10K | 20.00 | 81.00 | **43.40** | 48.10 |

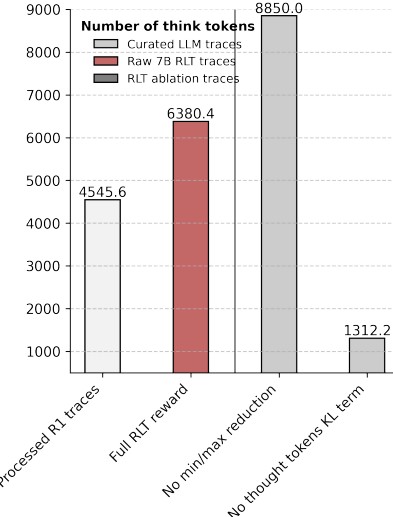

Figure 8: Average number of think tokens in the reasoning traces generated by our baseline R1 pipeline, Our 7B RLT, and other 7B teachers after ablating some of the key components in the RLT reward function.

### C.5 Student performance and scale of RLT dataset

We collected additional results by changing the amount of student distillation data, only focusing on the NuminaMath dataset [52]. We constructed new datasets with 1K and 10K samples using the AIME, MATH, and Olympiads subsets of NuminaMath from Li et al. [12], enabling us to directly compare with the available reasoning traces from our best baseline using postprocessed reasoning traces from DeepSeek R1, used for Bespoke-7B. As shown in the Table 13 above, we find that the performance of the RLT and R1 traces from the Bespoke pipeline positively scales with the amount of student training data. While we removed all science questions from STILL-2 present in our original data, we find performance on GPQA does not decrease substantially, evidencing positive transfer across different reasoning domains. We note these results are consistent with our previous random subsampling of the questions from Li et al. [12], with the RLT traces outperforming the postprocessed R1 traces across all tested data sizes and configurations.

## D Extended analysis

### D.1 RLT reward and R1 traces analysis

We analyze the empirical consequences of ablating the components in our RLT reward function, described in Section 3.3. In particular, we focus on the effects of removing two key components of our design. First, we examine ablating the think tokens KL reward $r^{\text{KL}}$, quantifying whether the

Table 14: Teachers ablated of individual reward components compared with RLTs and prior distillation pipelines on our main set of challenging reasoning tasks.

| Model | Data size | AIME 2024 | MATH 500 | GPQA Diamond | Overall |
|---|---|---|---|---|---|
| Qwen2.5-7B-Instruct | N.A. | 10.00 | 74.20 | 33.30 | 39.17 |
| Bespoke-7B (R1 traces) | 17K | 20.00 | 82.00 | 37.80 | 46.60 |
| Full RLT reward | 17K | **23.30** | **82.80** | **42.40** | **49.50** |
| No thought tokens KL term | 17K | 6.70 | 63.80 | 31.80 | 34.10 |
| No min/max reduction | 17K | **23.30** | 79.00 | 40.00 | 47.43 |

---

**Postprocessed R1 reasoning distillation example**

*Masked distillation input*

```
<|im_start|>system
<|im_start|>system
Your role as an assistant involves thoroughly exploring questions...
<|im_start|>user
Generate an executable Python function generated from the given prompt...
<|im_end|>
<|im_start|>assistant
```

*Distillation reasoning trace*

```
<|begin_of_thought|>
Okay, let's see. The problem is to find the minimum number of sheets needed to print N pages double-sided. Each sheet can hold two pages. So, I
↪  need to figure out how to calculate this based on N.
Hmm, let's think about some examples. Like the sample input is 5, output is 3. Because 5 divided by 2 is 2.5, which rounds up to 3. So maybe the
↪  general formula is to divide N by 2 and round up.
Wait, but wait. Let me check another example. Suppose N is 4. Then 4/2 is exactly 2 sheets. That makes sense. If N is 1, then 1/2 is 0.5, which
↪  rounds up to 1. So that's correct.
So the formula is simply ceiling(N / 2). But how to compute that in Python.
In Python, if you do integer division with //, but when you need to round up, you can do (N + 1) // 2. Let's see. For N=5: (5 +1)//2 =6//2=3.
↪  Correct. For N=4: (4+1)/2=5/2=2.5, integer division gives 2. Wait, no. Wait, 5//2 in Python is 2. So (4+1)//2 is 5//2=2. Which is correct.
↪  For N=1: (1+1)//2=1. Correct. So yes, the formula is (N +1) // 2.
So the function is straightforward. Take the input N, compute (N +1)//2, and output it.
Testing the sample input: input 5. (5+1)//2=6//2=3. Correct.
Another test case: N=1 → 1. N=2 → 1. N=3 →2. All correct.
So the code would be something like:
Read N from input, compute (N+1)//2, print the result.
So the Python code would be:
n = int(input())
print( (n +1) // 2 )
But wait, perhaps there's an edge case. Let's see. For N=0? But according to constraints, N is at least 1, so no problem.
Yes, so that's the solution. The code is simple. Just read N, calculate (N +1)//2, print.
<|end_of_thought|>
```

*Distillation solution*

```
<|begin_of_solution|>
To solve this problem, we need to determine the minimum number of sheets of paper required to print a document with N pages using double-sided
↪  printing, where each sheet can hold two pages.
### Approach
The key insight here is that each sheet of paper can hold two pages. Therefore, the minimum number of sheets required is determined by dividing
↪  the number of pages by 2 and rounding up to the nearest integer. This can be efficiently computed using integer arithmetic by adding 1 to
↪  the number of pages before performing integer division by 2. This approach works because it effectively rounds up the result of dividing N
↪  by 2.
### Solution Code
```python
n = int(input())
print((n + 1) // 2)
```
### Explanation
1. **Reading Input**: The input is read as an integer N, which represents the number of pages.
2. **Calculation**: The formula `(N + 1) // 2` is used to compute the minimum number of sheets. Adding 1 to N before division ensures that any
↪  remainder from the division results in rounding up to the next whole number.
3. **Output**: The result is printed, which is the minimum number of sheets needed.

This approach efficiently computes the result in constant time, O(1), making it very efficient even for the upper constraint limit.

<|end_of_solution|><|im_end|>
```

Figure 9: Full formatted distillation data sample using a postprocessed DeepSeek R1 reasoning trace from the Bespoke dataset [13].

think tokens $t_{o_i}$ themselves are interpretable logical continuations from the student's perspective as compared with the teacher's. Second, we examine ablating the min/max reductions terms, which serve to ensure the rewards do not forego any individual token, avoiding introducing bias to the reward values based on the solution length or the number of think tokens in the teacher's explanations.

For these experiments, we train entirely new 7B RLTs following the hyperparameters from Table 3 but setting $\lambda = 0$ (see Equation 5) or $\alpha = 0$ (see Equations 4 and 3), respectively for the thought tokens KL reward and min/max reduction ablations. We then construct student distillation datasets, using the same full set of starting question-solution pairs as our recent state-of-the-art baselines [12, 13] as considered in our main Section 4 experiments. We compare the generated datasets and students after

Masked distillation input

```
<|im_start|>system
<|im_start|>system
Your role as an assistant involves thoroughly exploring questions...
<|im_start|>user
Generate an executable Python function generated from the given prompt...
<|im_end|>
<|im_start|>assistant
```

Distillation reasoning trace

```
<|begin_of_thought|>
Okay, let's see. The problem is to find the minimum number of sheets needed to print N pages double-sided. Each sheet can hold two pages. So, I
↪  need to figure out how to calculate this based on N.

Hmm, let's think about some examples. Like the sample input is 5, output is 3. Because 5 divided by 2 is 2.5, which rounds up to 3. So maybe the
↪  general formula is to divide N by 2 and round up.

Wait, but wait. Let me check another example. Suppose N is 4. Then 4/2 is exactly 2 sheets. That makes sense. If N is 1, then 1/2 is 0.5, which
↪  rounds up to 1. So that's correct.

So the formula is simply ceiling(N / 2). But how to compute that in Python.

In Python, if you do integer division with //, but when you need to round up, you can do (N + 1) // 2. Let's see. For N=5: (5 +1)//2 =6//2=3.
↪  Correct. For N=4: (4+1)/2=5/2=2.5, integer division gives 2. Wait, no. Wait, 5/2 in Python is 2. So (4+1)//2 is 5/2=2. Which is correct.
↪  For N=1: (1+1)//2=1. Correct. So yes, the formula is (N +1) // 2.

So the function is straightforward. Take the input N, compute (N +1)//2, and output it.

Testing the sample input: input 5. (5+1)//2=6//2=3. Correct.

Another test case: N=1 → 1. N=2 → 1. N=3 →2. All correct.

So the code would be something like:

Read N from input, compute (N+1)//2, print the result.

So the Python code would be:
n = int(input())
print( (n +1) // 2 )
But wait, perhaps there's an edge case. Let's see. For N=0? But according to constraints, N is at least 1, so no problem.

Yes, so that's the solution. The code is simple. Just read N, calculate (N +1)//2, print.

<|end_of_thought|>
```

Distillation solution

```
<|begin_of_solution|>
To solve this problem, we need to determine the minimum number of sheets of paper required to print a document with N pages using double-sided
↪  printing, where each sheet can hold two pages.

### Approach
The key insight here is that each sheet of paper can hold two pages. Therefore, the minimum number of sheets required is determined by dividing
↪  the number of pages by 2 and rounding up to the nearest integer. This can be efficiently computed using integer arithmetic by adding 1 to
↪  the number of pages before performing integer division by 2. This approach works because it effectively rounds up the result of dividing N
↪  by 2.

### Solution Code
```python
n = int(input())
print((n + 1) // 2)
```

### Explanation
1. **Reading Input**: The input is read as an integer N, which represents the number of pages.
2. **Calculation**: The formula `(N + 1) // 2` is used to compute the minimum number of sheets. Adding 1 to N before division ensures that any
↪  remainder from the division results in rounding up to the next whole number.
3. **Output**: The result is printed, which is the minimum number of sheets needed.

This approach efficiently computes the result in constant time, O(1), making it very efficient even for the upper constraint limit.

<|end_of_solution|><|im_end|>
```

Figure 10: Full formatted distillation data sample using the raw reasoning trace from our 7B RLT.

distillation with these ablated teachers to our original 7B RLT trained with the full RLT reward and our strongest distillation baseline using postprocessed R1 reasoning traces [12, 13].

First, we focus on the effects of these ablations in terms of the traces' content as compared to our original RLT's traces and the postprocessed R1 pipeline. As shown in Figure 8, the length of the reasoning traces is greatly affected by our ablations. When looking at our original RLT with full rewards, the average length of the produced reasoning trace is 39% higher than the original curated R1 traces in the Bespoke dataset [13]. This is consistent with our analysis from Section 4.5 and further highlighted by the full examples shown in Figures 9 and 10, showing how our 7B RLT often includes alternative verification steps and approaches not considered by traditional pipelines that do not optimize directly for downstream distillation. Furthermore, ablating each term in our reward leads to the concrete unwarranted effects described in Section 3.3, allowing the RL optimization procedure to find "shortcuts" to maximize $r^{\text{RLT}}$ that hurt the quality of the reasoning traces. In particular, without the thought tokens KL reward $r^{\text{KL}}$, our RLT cannot differentiate between explanations that guide the student step-by-step and those that increase the solution's likelihood without a logical path that can be learned from. Thus, as shown in Figure 11, this ablation leads to a teacher that only learns to repeat the solution tokens themselves in its explanation to exploit the repetition tendency of pretrained student LMs, with the average length of its output reasoning traces dramatically dropping. Moreover, in our second ablation of the min/max reward reduction terms, the rewards become effectively biased by the length of the reasoning trace, leading the teacher to prefer long explanations only to reduce the influence on $r^{\text{KL}}$ of hard but necessary individual logical steps. As a consequence of this bias, the average number of think tokens in the teacher's reasoning traces almost doubles from

Masked distillation input

```
<|im_start|>system
<|im_start|>system
Your role as an assistant involves thoroughly exploring questions...
<|im_start|>user
Generate an executable Python function generated from the given prompt...
<|im_end|>
<|im_start|>assistant
```

Distillation reasoning trace

```
<|begin_of_thought|>
To solve this problem, we need to determine the minimum number of sheets of paper required to print a document with N pages using double-sided
↪   printing, where each sheet can hold two pages.
### Approach
The key insight here is that each sheet of paper can hold two pages. Therefore, the minimum number of sheets required is determined by dividing
↪   the number of pages by 2 and rounding up to the nearest integer. This can be efficiently computed using integer arithmetic by adding 1 to
↪   the number of pages before performing integer division by 2. This approach works because it effectively rounds up the result of dividing N
↪   by 2.
### Solution Code
```python
n = int(input())
print((n + 1) // 2)
```

### Explanation
1. **Reading Input**: The input is read as an integer N, which represents the number of pages.
2. **Calculation**: The formula `(N + 1) // 2` is used to compute the minimum number of sheets. Adding 1 to N before division ensures that any
↪   remainder from the division results in rounding up to the next whole number.
3. **Output**: The result is printed, which is the minimum number of sheets needed.

This approach efficiently computes the result in constant time, O(1), making it very efficient even for the upper constraint limit.

<|end_of_thought|>
```

Distillation solution

```
<|begin_of_solution|>
To solve this problem, we need to determine the minimum number of sheets of paper required to print a document with N pages using double-sided
↪   printing, where each sheet can hold two pages.
### Approach
The key insight here is that each sheet of paper can hold two pages. Therefore, the minimum number of sheets required is determined by dividing
↪   the number of pages by 2 and rounding up to the nearest integer. This can be efficiently computed using integer arithmetic by adding 1 to
↪   the number of pages before performing integer division by 2. This approach works because it effectively rounds up the result of dividing N
↪   by 2.
### Solution Code
```python
n = int(input())
print((n + 1) // 2)
```

### Explanation
1. **Reading Input**: The input is read as an integer N, which represents the number of pages.
2. **Calculation**: The formula `(N + 1) // 2` is used to compute the minimum number of sheets. Adding 1 to N before division ensures that any
↪   remainder from the division results in rounding up to the next whole number.
3. **Output**: The result is printed, which is the minimum number of sheets needed.

This approach efficiently computes the result in constant time, O(1), making it very efficient even for the upper constraint limit.

<|end_of_solution|><|im_end|>
```

Figure 11: Full formatted distillation data sample using a reasoning trace collected after training our teacher ablated from the thought tokens KL reward $r^{\text{KL}}$ in the RLT rewards defined in Equation 5.

the postprocessed R1 traces and, as shown in Figure 12, their content starts including many additional unnecessary steps that are just semantical repetitions of each other as learning progresses.

Then, we also quantify and compare the effects that each of our ablations has on downstream student performance. As shown in Table 14, the disruptive effects of ablating the thought tokens KL reward entirely reflect on the capabilities of the learned students, with their performance being lower than even the original Qwen-7B model they are fine-tuned from. This result validates our reward design, showing how regularizing for the reasoning traces to be "natural" continuations from the student's own perspective is of key importance for effective distillation. On the other hand, ablating the min/max reward reduction terms produces a more moderate reduction in performance, with the new output traces of this 7B teacher still remarkably outperforming our strongest baseline pipeline using the R1 LM with orders of magnitude more parameters. However, we note that by preventing the increase in the lengths of the reasoning traces with the min/max reduction terms, our full RLT reward also yields faster training, distillation dataset generation, and student fine-tuning, with non-trivial benefits contributing to our framework's efficiency.

# E  Extended discussion and limitations

## E.1  Traditional RL and the importance of overcoming reward sparsity

In traditional RL for robotics, neural network policies can learn to solve tasks from random initializations thanks to being guided by dense reward functions designed and tuned to provide domain-specific guidance on the level of progress [53, 54]. Without this guidance, they would be faced with an infeasible exploration challenge made exponentially more difficult by the task horizon [55]. This is

because dense rewards allow the policy gradient optimization to rank the relative progress obtained across its initial suboptimal actions, allowing the policy to bootstrap from partial solutions and extrapolate far beyond its initial capabilities. However, in the context of traditional LM reasoning, as correctness-based rewards are inherently sparse, this extrapolation is not possible. In particular, if an LM is too weak to provide the optimal solution for a task, all its rewards (and, thus, the policy gradient) will be zero. In contrast, we note this limitation would not apply to our RLT, which can make use of a dense reward function and obtain a learning signal to iteratively improve even when all sampled initial explanations are suboptimal, as with the traditional RL framework.

## E.2 Limitations and unexplored directions

This work's purpose was to introduce a new class of Reinforcement-Learned Teachers designed to avoid the exploration challenge of sparse rewards and align the optimization of RL-trained LMs with the test-time goal of downstream distillation. However, there are still several limitations and improvements which we hope will be tackled in future extensions. First, to output explanations, the RLT framework relies on access to the ground-truth solutions. Hence, when used with datasets and domains where this information might not be available or not be practical to recover without querying LMs, small RLTs might have to again rely on larger models, even though still to a lesser extent than prior distillation pipelines. Similarly, as described in Section 4.1 and Appendix A, our current training recipe does not exclusively involve RL, with an initial phase to familiarize our small model with its new teaching format and role, again relying on some level of initial access to pre-collected reasoning examples that can be used accordingly. We also note that RLT training makes use of an additional student model for computing the rewards. In practice, however, we note the downsides from this were minimal, as simple parameter offloading removed any potential memory burden of having this model on GPU memory during backpropagation, and actual training time was dominated ($> 90\%$) by the costs of long-context autoregressive generation with our computational setup. Recent work has shown early promise in overcoming this inherent bottleneck of RL methods by grounding the model's reasoning process away from self-generated tokens, and using separate spaces such as diffusion [56]. Yet, while this complementary approach can be more efficient, it has yet to have the same impact as RL reasoning at larger scales.

As described in Section 6, we also did not explore the potential of sharing the teacher and student roles with the same optimized model, nor concurrent training of the two. This leaves two open questions yet unanswered: whether there can be an effective transfer between the two roles, potentially improving training efficiency by harnessing the relatedness of their objectives, and whether RL can devise online curricula tailored to the student's specific learning dynamics. Furthermore, due to computational constraints, we had to limit the maximum context size during our RL-training and student distillation phases to 16384, half the maximum context available with Qwen-based models. For the same reason, the considered RLTs only comprised small, inexpensive 7B models and did not consider further scaling. Breaking these constraints, thus, also remains another outstanding immediate direction for future improvements and further pushing the capabilities of our framework. Lastly, this work did not explore increasing the number and breadth of starting question-solution pairs used for obtaining distillation datasets, beyond the ones considered in prior work that made use of much more expensive pipelines. This leaves the potential extent of another key feature of our method unexplored, as with our small 7B RLT, it could be much more feasible to cheaply collect data for student distillation that matches the level of scale and efficacy of closed-source state-of-the-art data sources such as the one from Guo et al. [4].

## E.3 Broader impact

Our work introduced a new class of models that enable small LMs to generate better synthetic distillation datasets beyond prior, much more expensive pipelines. Rather than introducing a new application, this work's contribution was foundational in nature, thus, its broader implications are bound to the effect of improving the capabilities and democratizing the training of large language models. To this end, as the accessibility and capabilities of LMs improve, there is an increasing chance of misuse for potentially harmful goals, such as influencing public opinion or obtaining access to sensitive information. Moreover, with increasing demand, the carbon footprint of LMs and their potential social impact might become increasingly relevant. However, we believe these risks are currently offset by the potential upsides of advancing the AI field, such as empowering humanity to better face upcoming environmental and economic challenges.

## Min/max reduction reward ablation distillation example

```
<|begin_of_thought|>
Okay, I need to solve this problem where Takahashi wants to print N pages double-sided, with each sheet holding two pages. The task is to find
↪  the minimum number of sheets required. Let's think through this step by step.
So, the problem is about packing pages into sheets. Each sheet can have two pages. But since it's double-sided, maybe the way the pages are
↪  arranged affects how they fit. Wait, no, no, double-sided printing just means that two pages are printed on the same sheet, front and back. So
↪  for example, if I have page 1 on the front and page 2 on the back, that's one sheet. But if there's an odd number of pages, the last sheet
↪  will have only one page. For example, if N is 5, then the sample input shows 3 sheets. So if there's see: 1-2 on first sheet, 3-4 on second, 5 on
↪  third. So each sheet except possibly the last can have two pages.
Hmm, so the approach is to calculate how many pairs we can make. For N pages, the number of pairs is N//2. But if N is odd, there's an extra
↪  page. So the total number of sheets would be (N // 2) + (1 if N % 2 != 0 else 0). Wait, but in the sample input, N is 5. 5//2 is 2, and 5%2
↪  is 1, so 2+1=3. Which matches the sample output. So that's correct.
Wait, but wait. Let's check another example. If N is 4. Then 4//2 is 2, and 4%2 is 0. So 2 + 0 = 2. Which would be two sheets. Each sheet has
↪  two pages. Like 1-2 on first, 3-4 on second. So that's correct.
Another example: N=3. 3//2 is 1, 3%2 is 1. So 1+1=2. So two sheets. Like 1-2 on first, 3 on the second. Correct.
What about N=1? Then 1//2 is 0, 1%2=1. So 0+1=1. So one sheet. But that's impossible because you can't print a single page on a double-sided
↪  sheet. Wait, but according to the problem statement, the constraints say that N is at least 1. But the problem says that two pages can be
↪  printed on a sheet. So for N=1, the answer is 1? Because there's only one page, so you can't do double-sided. Wait, but wait, the problem
↪  says "double-sided, where two pages of data can be printed on one sheet of paper." So if there's an odd number of pages, the last sheet will
↪  have one page. But in that case, that's allowed. Wait, but for a single page, you can't print double-sided. So maybe the problem's
↪  constraints don't have N=1, but according to the constraints given here, 1 <= N <= 100. So the problem allows N=1. But according to the
↪  problem statement, the answer for N=1 is 1? But that's impossible. Wait, perhaps I'm misunderstanding the problem.
Wait, the problem says each sheet can hold two pages. So if there's an odd number of pages, the last sheet has one page. But the problem's
↪  sample input is 5, which works out to 3 sheets. So maybe when N is odd, the formula is (N+1)/2. Wait, 5+1 is 6, 6/2 is 3. Which is
↪  correct. For N=3, 3+1=4, 4//2=2. Correct. For N=1, 1+1=2, 2//2=1. So that's correct. Wait, 1+1=2, 2//2=1. So the formula is (N + 1) // 2.
↪  Let's test that.
Wait, for N=1: (1+1)//2 = 1. Which is correct. For N=2: (2+1)/2=1.5, integer division is 1. Which is correct. For N=3: (3+1)/2=2. Correct. N=4:
↪  5//2=2.5→2. Correct. N=5: 3. Correct. So this formula seems to work.
Wait, but wait. Let's see. For N=1, if we use (N+1)//2, that's (1+1)//2=1. So yes. So the formula is (N +1)//2. But wait, in the problem
↪  statement, the sample input 5 gives output 3, which is (5+1)//2=3. So that's correct. Because adding 1 gives
↪  even number of pages, which can be divided by 2, and then integer division by 2 gives the correct number of sheets.
Wait, but let me think again. For N=1, the formula gives 1. So that's correct. For N=2, (2+1)//2=1.5→1. Which is correct. So yes, this formula
↪  works.
Wait, but why would (N+1)/2 be the correct formula? Because when you have N pages, each sheet holds two pages, except possibly the last one. So
↪  the total number of sheets is the number of pairs (each pair is two pages) plus one if there's an odd number. So the number of pairs is
↪  N//2, and if N is odd, there's an extra sheet. So the total is (N//2) + (1 if N %2 !=0 else 0). Which can be written as (N+1)//2.
Yes. Because for N even, (N+1)/2 is (N/2) +0.5 → but integer division would be N/2. For example, N=4: (4+1)/2=2.5→2. Which is correct. For N=5:
↪  (5+1)/2=3. So that's correct.
So the formula is (N +1) // 2.
Wait, but wait. Let's test N=1. (1+1)//2=1. Correct. N=2: (2+1)/2=1.5→1. Correct. N=3: (3+1)/2=2. Correct. N=4: 2. Correct. N=5:3. Correct.
So this formula works.
So the problem reduces to calculating (N + 1) // 2.
But wait, the input is N, which is between 1 and 100. So the function needs to read N from standard input, compute (N+1)//2, and output that.
Wait, but wait. Let's take N=1. The sample input is 5, which gives output 3. For N=1, the output would be 1. But according to the problem
↪  statement, the minimal number of sheets. So for N=1, you need 1 sheet. Because even though you can't fill a full sheet, you just have to use
↪  one sheet. So the formula works.
So the code is straightforward. Read N, compute (N+1)//2, print that.
Wait, but let me check another example. Let's say N=6. (6+1)//2=3.5→3. Wait, 6//2=3. Correct. So that's correct.
Wait, but wait. Wait, no. Wait, (6+1)//2 is 7//2=3. Wait, 7 divided by 2 is 3.5, integer division is 3. So (6+1)//2 is 3.5 →3. So yes. So 6
↪  pages would require 3 sheets. Each sheet has two pages. So 1-2, 3-4, 5-6. So three sheets. Correct.
So the code is as simple as reading N, then compute the ceiling of N divided by 2. Which is (N +1)//2.
So the Python function would be:
Read N from input. Then compute (N +1) // 2. Print that result.
Wait, but wait. Let's test N=1 again. (1+1)//2=1. Correct.
Yes. So the code is straightforward.
So the code would be:
Read N as integer from input. Then print (N + 1) // 2.
But wait, wait. Let's check N=1. (1+1)//2=1. Correct. For N=2: (2+1)//2=1. So 1 sheet. Which is correct. For N=3: 2 sheets. Correct. For N=4: 2.
↪  Correct. For N=5:3. Correct. For N=6:3. Correct.
So the code is correct.
Therefore, the Python function is:
Read N from input, then compute (N + 1) // 2 and print it.
So the code would be:
n = int(input())
print((n + 1) // 2)
Wait, but wait. Let me think again. For N=1, the answer is 1. For N=2, 1. So that's correct. So the code is correct.
But wait, the problem says that N is the number of pages. So that's correct. So the code is as simple as that.
I think that's it. So the answer is to compute (N +1) // 2.
<|end_of_thought|>
```

```
<|begin_of_solution|>
To solve this problem, we need to determine the minimum number of sheets of paper required to print a document with N pages using double-sided
↪  printing, where each sheet can hold two pages.
### Approach
The key insight here is that each sheet of paper can hold two pages. Therefore, the minimum number of sheets required is determined by dividing
↪  the number of pages by 2 and rounding up to the nearest integer. This can be efficiently computed using integer arithmetic by adding 1 to
↪  the number of pages before performing integer division by 2. This approach works because it effectively rounds up the result of dividing N
↪  by 2.
### Solution Code
```python
n = int(input())
print((n + 1) // 2)
```

### Explanation
1. **Reading Input**: The input is read as an integer N, which represents the number of pages.
2. **Calculation**: The formula `(N + 1) // 2` is used to compute the minimum number of sheets. Adding 1 to N before division ensures that any
↪  remainder from the division results in rounding up to the next whole number.
3. **Output**: The result is printed, which is the minimum number of sheets needed.
This approach efficiently computes the result in constant time, O(1), making it very efficient even for the upper constraint limit.
<|end_of_solution|><|im_end|>
```

Figure 12: Full formatted distillation data sample using a reasoning trace collected after training our teacher ablated from the min/max reduction term in the RLT rewards components defined in Equations 4 and 3. System prompt and Assistant message omitted for space.

