# OpenReview forum: "Reinforcement Learning Teachers of Test Time Scaling"
_NeurIPS.cc/2025/Conference — NeurIPS 2025 poster_

### Official Review · Reviewer_YMHP · 2025-06-29

**Clarity:** 3
**Significance:** 3
**Originality:** 3
**Rating:** 6
**Confidence:** 4

**Summary:**

The goal of this paper is to create better teachers for distillation, specifically distillation of long-chain-of-thought "reasoning" models. The core concept is to only have the teacher rationalize existing answers to given questions, rather than come up with new answers to existing questions from scratch or come up with wholly new questions. To train LLMs to be better teachers for distillation, the authors add two objectives to RL training for the teacher. The first loss rewards the teacher for providing explanations (chains-of-thought) that increase the student's log-probability of the solution tokens. The 2nd loss rewards the teacher for providing explanations (chains-of-thoughts) that also have a high probability under the student policy. A series of experiments shows that that models trained with these losses make better trainers for distillation and better starting points for RL training.

**Questions:**

Please see the "Weaknesses" section for my questions.  I think an ablation study for the impact of the two losses would be the most useful. I think the other experiments are likely too compute intensive to be done during the rebuttal period, thought I think the scaling experiment is still useful to see, especially with a dataset like NuminaMath-1.5. I'm interested in seeing, for example, how the performance of a student distilled from the teacher changes with / without the RLT teacher over differing amounts of NuminaMath data (perhaps 1k, 10k, 100k, 1M). Finally, see my comments about the related work.

**Ethical Concerns:**

["NO or VERY MINOR ethics concerns only"]

**Final Justification:**

This paper makes an important contribution to data-centric research involving distillation. This is an important part of many data pipelines for open LLMs. The proposed approach is a technically ambitious one (of which simpler variants have been previously attempted) to sort of "meta-learn" how to improve the data (e.g. by training the teacher). This is pretty hard to do, and the authors propose a method that clearly seems to work, and is relatively elegant.

I had questions about an ablation, but the authors have done the ablation. I also had an issue with a missing discussion of papers that are attempting the same meta-learning approach, but the authors have also addressed this. The remaining issue is that the authors only experiment on smaller data scales, but I think this is unavoidable in the academic setting.

I have read the other reviews. I am not convinced by their weaknesses — some of them either seem irrelevant to how distillation is used in practice (e.g. "cases where teacher is smaller than student") or miss the aim of the paper (e.g. "method only improves teacher instead of student") or very out of scope (e.g. "method does not allow solving problems with unknown answers").

I think there are interesting scientific insights in this paper regarding how reinforcement learning on LLMs works on tasks with a very complex reward like learning to give better explanations, and there are practical applications here. Generally, I think this will be quite useful to the community and many people will be interested in the idea.

**Limitations:**

There wasn't much discussion of limitations, but I also don't think anything very specific was needed here.

**Paper Formatting Concerns:**

None.

**Quality:**

3

**Strengths And Weaknesses:**

# Strengths
- The central idea of the paper is well-motivated and implemented elegantly with two easy-to-compute losses that both make sense at a glance.
- The experimental results are strong; it's clear the RLT-trained teacher is a better teacher than the baselines across multiple benchmarks.
- Given the interest indicated by the community in distilling reasoning models, I think the technique described here + the strong results could be built on by others.
- I thought the out of distribution analysis was _really interesting_ and backs up one of the core motivations of the paper, that focusing on making better teachers (provided verifiable answers are present) is maybe a better use of compute than doing RL with sparse rewards.

# Weaknesses
- I didn't see an ablation study for the two losses. Do we really need both of them?
- I think distillation is something very data dependent and conclusions might change a lot with scale and where questions + answers are being sourced from. I didn't see an analysis of this in the paper.
- There are no comparisons with models that are designed for distillation, like Nvidia's Nemotron-4. But this might be beyond the compute constraints of the authors.
- The idea of the paper can be summed up as "do RL to make better teachers for distillation". But the Related Works section includes few discussions or mentions of papers that are working on "better teachers". Examples are [Montessori-Instruct](https://arxiv.org/abs/2410.14208) and [DataEnvGym](https://arxiv.org/abs/2410.06215), which both have similar motivations but different ways of achieving them. I think this paper is different enough that it doesn't need to justify it's novelty, but it should situate itself relative to prior work better.

---

> ### Author Rebuttal · Authors · 2025-07-30
>
> # Reviewer 4 YMHP
>
> We would like to thank reviewer YMHP for their feedback and the time they dedicated to our review. We worked to address each of their comments and hope they will not hesitate to let us know of any further suggestions or questions.
>
> **responses**
>
> > 1) I didn't see an ablation study for the two losses. [...] I think an ablation study for the impact of the two losses would be the most useful.
>
> We would like to clarify that we did perform an in-depth ablation of our reward terms in Section D of our Appendix.
> 1. First, we examined ablating the think tokens KL reward term $r^{KL}$, quantifying whether the think tokens $t_{o_i}$ themselves are interpretable logical continuations from the student's perspective as compared with the teacher's.
> 2. Second, we examined ablating the min/max reductions terms, which serve to ensure the rewards do not forego any individual token, as detailed in Section 3.3.
>
> We provided these results in Table 7 and summarize them again in the Table below for easier reference:
>
> |Evaluated model|AIME 2024|MATH 500|GPQA Diamond|Overall|
> |-|-|-|-|-|
> |Qwen2.5-7B-Instruct|10.0|74.2|33.3|39.2|
> |Bespoke-7B(R1 traces)|20.0|82.0|37.8|46.6|
> |Full RLT reward|23.3|82.8|42.4|49.5|
> |No thought tokens KL term $r^{KL}$|6.7|63.8|31.8|34.1|
> |No min/max reward reduction|23.3|79.0|40.0|47.4|
>
> Furthermore, beyond performance, in Figures 8-12, we also compared the lengths of the produced teacher explanations and provided several qualitative, concrete examples showing the effects of each reward term. As detailed in Appendix D.1, ablating $r^{KL}$ leads to a teacher that only learns to repeat the solution tokens themselves in its explanation to exploit the repetition tendency of pretrained student LMs, with the distillation performance and average length of its output reasoning traces dramatically dropping. Furthermore, ablating the min/max reward reduction terms makes the rewards effectively biased by the length of the reasoning trace. While this second ablation only minimally influences performance, it makes the number of think tokens in the teacher's reasoning traces almost double, unnecessarily increasing the downstream student distillation cost.
>
> Following the reviewer’s feedback, we extended the end of Section 3.3 to reference more explicitly the presence of this set of important results validating our reward design, and avoid readers potentially missing them. In particular, we modified lines 198-199 to:
>
> “For further discussion, we refer to Appendix D, where we empirically analyze and validate each design choice in our reward function in terms of final performance (Table 7) and concrete qualitative differences of the resulting explanations (Figures 8-11).”
>
> We note that we did not consider ablating $r^{SS}$ to devolve our compute resources to the other experiments, as without this term our RLT would simply collapse to the student’s initial distribution to minimize $r^{KL}$ with no incentive to make its explanations useful. In fact, this is precisely what we also observed in early runs where the weighting coefficient $\lambda$ of $r^{KL}$ was too high. However, we hope the reviewer will not hesitate to let us know in case they believe we should still collect these results and add them to our ablation study in future revisions.
>
> > 2) Distillation is something very data dependent and conclusions might change a lot with scale and where questions + answers are being sourced from [...] I'm interested in seeing, for example, how the performance of a student distilled from the teacher changes.with / without the RLT teacher over differing amounts of NuminaMath data.
>
> We definitely agree with the reviewer that distillation performance can be very dependent on the scale and quality of the initial question dataset. As described in Sections 4.1 and 4.2, this is precisely the reason that we distilled our students using the same starting set of 17K question-solution pairs from the recent state-of-the-art distillation baselines used in our comparison [1, 2]. This allowed us to remove potential confounding factors other than the quality of the reasoning traces between the ones generated from our RLT and much more expensive LLMs such as DeepSeek R1 [3] (we note that we also matched our baselines’ exact distillation hyperparameters and evaluation code rather than making our own).
>
> Following the reviewer’s feedback, we collected and added new results in Appendix C, changing the amount of student distillation data, focusing on the NuminaMath dataset. While we did not have time to experiment with hundreds of thousands of samples, we constructed new datasets with 1K, and 10K samples from the AIME, MATH, and Olympiads subsets of NuminaMath as done in [1], enabling us to directly compare with the available reasoning traces from our best baseline using postprocessed reasoning traces from DeepSeek R1 (i.e, Bespoke-7B [2]):
>
> |Evaluated model|Distillation data|AIME 2024|MATH 500|GPQA Diamond|Overall|
> |-|-|-|-|-|-|
> |Qwen2.5-7B-Instruct|N/A|10.0|74.2|33.3|39.2|
> |Bespoke-7B-1K|1K subset from [1]|13.3|80.0|33.8|42.4|
> |Bespoke-7B|17K data from [1]|20.0|82.0|37.8|46.6|
> |Bespoke-7B(1K NuminaMath)|1K subset from NuminaMath|16.7|79.0|33.3|43.0|
> |Bespoke-7B(10K NuminaMath)|10K subset from NuminaMath|16.7|79.6|40.9|45.7|
> |RLT-7B-1K|1K subset from [1]|20.0|80.4|41.9|47.4|
> |RLT-7B|17K data from [1]|23.3|82.8|42.4|49.5|
> |RLT-7B(1K NuminaMath)|1K subset from NuminaMath|16.7|80.4|40.4|45.8|
> |RLT-7B(10K NuminaMath)|10K subset from NuminaMath|20.0|81.0|43.4|48.1|
>
> As summarized in the Table above, we find that the performance of the RLT and R1 traces from the Bespoke pipeline positively scales with the amount of student training data. While we removed all science questions from STILL-2 present in our original data [4], we find performance on GPQA does not decrease substantially, evidencing positive transfer across different reasoning domains. We note these results are consistent with our previous random subsampling of the questions from [1] (from Table 1 of our paper), with the RLT traces outperforming the postprocessed R1 traces across all tested data sizes and configurations.
>
> We hope these new results will provide further insights to our future readers regarding the scalability and robustness of our new framework.
>
> > 3) The Related Works section includes few discussions or mentions of papers that are working on "better teachers". Examples are Montessori-Instruct and DataEnvGym [...] I think this paper is different enough that it doesn't need to justify it's novelty, but it should situate itself relative to prior work better.
>
> Following the reviewer’s feedback, we extended our Related Works section to include prior methods working toward similar goals of designing/training “autonomous data generation agents,”  including the two referenced works [5, 6] and others such as [7]. We made sure to better contextualize the contributions of our paper in relation to these papers in terms of its different methodology and its application to the domain of LM reasoning.
>
> **references**
>
> [1] Li, Dacheng, et al. "LLMs Can Easily Learn to Reason from Demonstrations Structure, not content, is what matters!."
>
> [2] Bespoke Labs. Bespoke-stratos: The unreasonable effectiveness of reasoning distillation.
>
> [3] Guo, Daya, et al. "Deepseek-r1: Incentivizing reasoning capability in llms via reinforcement learning."
>
> [4] Min, Yingqian, et al. "Imitate, explore, and self-improve: A reproduction report on slow-thinking reasoning systems."
>
> [5] Li, Xiaochuan, Zichun Yu, and Chenyan Xiong. "Montessori-instruct: Generate influential training data tailored for student learning."
>
> [6] Khan, Zaid, et al. "Dataenvgym: Data generation agents in teacher environments with student feedback."
>
> [7] Ning, Xuefei, et al. "Can LLMs learn by teaching for better reasoning? A preliminary study."

---

> > ### Comment · Reviewer_YMHP · 2025-08-05
> >
> > Thank you for the detailed and high-quality response! I found the ablation particularly interesting, and think for a potential arXiv version (without page limits) this should be moved to the main section of the paper since it also provides concrete intuition for how the teacher behaves under training. I will increase my score.

---

### Official Review · Reviewer_UdmA · 2025-07-01

**Clarity:** 2
**Significance:** 3
**Originality:** 3
**Rating:** 4
**Confidence:** 3

**Summary:**

This paper introduces Reinforcement-Learned Teachers (RLTs), a new framework for training language models to be effective teachers for knowledge distillation. Instead of conventional approach of training models to solve problems from scratch, the RLT framework tasks the teacher model to generate detailed explanations for problems given the solution. The authors demonstrate that a 7B RLT can produce raw reasoning traces that are more effective for distilling student models than the traces from much larger LMs (e.g., DeepSeek-R1, Gemini). The paper shows the effectiveness of RLTs for distilling students of various sizes, for cold-starting traditional RL, and for zero-shot transfer to new reasoning tasks.

**Questions:**

1. The analysis of cost: Could you provide a more direct comparison of the total computational cost (e.g., in GPU-hours) required to (a) train a 7B RLT and distill to a student, versus (b) generating a 17K dataset with a large model like DeepSeek-R1 and then distilling?
2. Change one student to diverse students: Recent work [1] provides empirical evidence that teaching a diverse set of students is more effective for improving the teacher. Have you ever considered changing the single student to diverse students(e.g. different model sizes or from different model families)? This can be implemented by either teaching a diverse group of students concurrently or by teaching different students at different training stages.
3. The scalability of the RLT framework: Have you ever tried the training of larger RLTs, for example, a 14B RLT? Does the effectiveness of the RLT framework continue to scale with the teacher's model size?
4. Why is $r^{KL}$ computed under the RLT format of the teacher? Since KL-Divergence naturally exists for RLT format and regular QA format.

[1] Can LLMs Learn by Teaching for Better Reasoning? A Preliminary Study

**Ethical Concerns:**

["NO or VERY MINOR ethics concerns only"]

**Final Justification:**

RLTs offer a well-motivated way to distill reasoning traces by letting the teacher model *reason with answer*. The rebuttal resolves my main concerns on cost analysis, writing clarity and figure notions. Although the scaling-up experiments are not done, a scaling-down experiments provide acceptable orcle of the scaling behavior. Thus, I maintain positive and give boarderline accept for the paper. The addtional scaling-up experiment is still advised for the final version.

**Limitations:**

Yes, the authors have addressed the limitations of their work. In Section 6 ("Discussion and extensions"), they acknowledge that their research is an initial exploration of the RLT framework and they propose several avenues for future work.

**Quality:**

3

**Strengths And Weaknesses:**

**Strengths**

1. **Innovative and intuitive framework.** The paper introduces an innovative and intuitive framework that reframes the conventional approach to training reasoning models. The core idea is training a RLT to create instructive explanations given question and its solution, instead of solving a question from scratch. This cleverly avoids the exploration challenge in traditional RL approaches. Besides, the intuition that explaining knowledge is more important than solving questions for real-world teachers provides a strong conceptual grounding for this work.
2. **Sufficient and compelling empirical results.** The paper shows strong results in many experiments. These experiments support the paper's claims. It demonstrates that raw outputs from a 7B RLT are more effective for distillation than processed traces from larger models, providing a superior cold-start for subsequent RL training and enabling zero-shot generalization to new domains. The authors also validate their method by showing a strong correlation between their reward scores and the performance of students trained on the corresponding data.

**Weakness**

1. **Lack of cost analysis.** The paper highlights efficiency gains by avoiding distillation from large models like deepseek-r1. However, the RLT training loop itself seems computationally intensive, requiring initial SFT, online generation from the teacher and forward passes through the student to calculate rewards for every sample and finally the distallitation process. A clear cost analysis is missing.
2. **Lack of analysis of the scaling behavior of the RLT.** The paper shows that a small (7B) RLT can effectively teach a larger (32B) student. However, it does not explore the scaling behavior of the teacher model.
3. **Minor:**
   1. Writing style. Some sentences are too long to easily comprehend. The author is recommended to break these down for better and clearer understanding.
   2. Figure clarity. In Figure 1, unclear whether it is the student size or teacher size

---

> ### Author Rebuttal · Authors · 2025-07-30
>
> We would like to thank reviewer UdmA for their feedback and the time they dedicated to our review. We worked to address each of their comments and hope they will not hesitate to let us know of any further suggestions or questions.
>
> **responses**
>
> > 1) A clear cost analysis is missing [W1]. Could you provide a more direct comparison of the total computational cost [...]? [Q1]
>
> Following the reviewer’s suggestion, we added a subsection to Appendix A, reporting the GPU cost of training and collecting distillation data with our RLT (recorded using a single H100 node with 8 GPUs, see Appendix A.1 for full hardware specs). We compared these times with an estimated minimum cost of traditional RL post-training and data collection using the DeepSeek R1 model. To compute this estimate, we used official figures for the GPUs used to train this DeepSeek model (2,048 NVIDIA H800 GPUs [1, 2]) and online communication from the DeepSeek team. Moreover, we also calculated a lower bound of the GPU hours for generating the distillation dataset by multiplying the minimum amount of GPU resources required to host DeepSeek R1 with half precision and the “roof inference speed” for generating each completion in our dataset, without taking into account slowdowns from the growing context size.
>
> However, as we do not have the resources to train/host DeepSeek R1 ourselves, we note that both these estimates should be considered large underestimations (we did not even consider time/resources for multiple generations, filtering, and post-processing stages conducted in our baselines [3, 4]). We provide a summary of this cost comparison in the Table below:
>
> |Model/GPU-hours|Training(GPU model)|Data generation(GPU model)|
> |-|-|-|
> |DeepSeek R1|>688000(H800)|>1067(H100)|
> |7B RLT teacher|280.4(H100)|6.7(H100)|
>
> We would also like to note that the training time of RLTs is largely dominated by the autoregressive generation loop during the RL phase, with the initial SFT and the extra forward passes through our student network adding a negligible amount of compute cost. This is also precisely the case in the traditional RL reasoning framework, as autoregressive generation comprises a large number of forward passes up to the maximum output context length specified for each single batch sample (16384 in our case).
>
> We hope that adding this new direct comparison and analysis to our paper will help better contextualize the trivial overheads and great potential of our new RLT framework to cut future requirements for training and distilling RL reasoning models.
>
> > 2) Recent work [5] provides empirical evidence that teaching a diverse set of students is more effective [...] This can be implemented by either teaching a diverse group of students concurrently or by teaching different students at different training stages. [Q2]:
>
> Following the reviewer’s suggestions, we added a direct reference to [5] and collected two new sets of experiments in Appendix D to increase student diversity for computing the RLT rewards:
> 1. First, we considered using both the 7B and the 32B students, averaging their probabilities, thus introducing student diversity into the RLT reward.
> 2. Second, we considered conducting training in two stages, each optimizing both teacher and student for half the total number of optimization steps. In particular, we paused the teacher RL optimization midway, distilled a student with this preliminary RLT’s explanations, and resumed our teacher optimization for the remaining steps with this updated student to compute the RLT reward.
>
> We note that this second extension has the added benefit of better aligning the teacher rewards to the student’s learning dynamics, and can be considered an initial step toward online optimization of both models (a future direction outlined in Section 6, lines 361-363).
>
> |Evaluated model|Teacher|AIME 2024|MATH 500|GPQA|Overall|
> |-|-|-|-|-|-|
> |Qwen2.5-7B-Instruct|N/A|10.0|74.2|33.3|39.2|
> |Bespoke-7B|DeepSeek R1|20.0|82.0|37.8|46.6|
> |RLT-7B|RLT teacher|23.3|82.8|42.4|49.5|
> |RLT-7B(7B + 32B $r^{RLT}$)|RLT teacher(7B + 32B $r^{RLT}$)|23.3|83.8|41.9|49.7|
> |RLT-7B(2-stage teacher)|RLT teacher(2-stage training)|26.7|84.0|41.4|50.7|
>
> As summarized in the Table above, both these extensions provide some moderate improvements to the RLT distillation performance. However, we also note that these came at a substantial additional training cost, taking approximately 1.4x and 1.8x more GPU hours than our original pipeline.
>
> > 3) The paper [...] does not explore the scaling behavior of the teacher model. [W2] Have you ever tried the training of larger RLTs, for example, a 14B RLT? [Q3]
>
> As detailed in our previous response, we ran all our experiments for a 7B RLT using a single H100 node. However, as the cost of training and deploying LLMs does not scale linearly with the number of parameters, our same experiments with a 14B RLT would have taken several weeks and required expensive multi-node setups, beyond the project's allocated resources.
>
> Thus, to still try to address the reviewer’s feedback, we instead focused on decreasing the size of the RLT teacher down to 3B parameters. We ran these experiments and added these new results to Appendix D:
>
> |Evaluated model|Teacher|AIME 2024|MATH 500|GPQA|Overall|
> |-|-|-|-|-|-|
> |Qwen2.5-7B-Instruct|N/A|10.0|74.2|33.3|39.2|
> |RLT-7B|RLT teacher(7B)|23.3|82.8|42.4|49.5|
> |RLT-7B(3B teacher)|RLT teacher(3B)|20.0|80.6|38.9|46.5|
> |Qwen2.5-32B-Instruct|N/A|26.7|84.0|49.0|53.2|
> |RLT-32B|RLT teacher(7B)|66.7|93.4|59.6|73.2|
> |RLT-32B(3B teacher)|RLT teacher(3B)|46.7|91.4|53.5|63.9|
>
> As summarized in the Table above, we indeed confirm that larger teachers yield better explanations and downstream results. However, most of the performance gap between our 3B and 7B teachers appears to occur only when distilling the 32B student LM, where the mismatch between teacher and student capabilities tends to the extreme. However, even in this case where the student is over 10x larger than the teacher, our 3B RLT is still able to provide considerable improvements to the initial Qwen 32B's performance.
>
> We believe these results further validate how the RLT framework enables even small, inexpensive models to be effective teachers, but also confirm that larger teacher models are indeed able to provide better explanations, highlighting the potential future implications of scaling with more training compute.
>
> > 4) Why is $r^{KL}$ computed under the RLT format of the teacher? [Q4]
>
> The $r^{KL}$ reward term is computed with the KL divergence over the distribution of think tokens outputted by our RLT during training (thus, under the RLT’s format with both the question and solution in context) and the student’s distribution over the same think tokens (under the distillation format with only the question in context). The purpose of $r^{KL}$ is to ensure that the think tokens in the explanations outputted by the teacher are interpretable logical continuations from the student’s perspective.
>
> Instead, if $r^{KL}$ was computed using the same student format also for our teacher (thus, seeing only the question in context), there would be no regularization preventing our teacher from outputting explanations simply copying and repeating the solution tokens to increase likelihood with the RLT format. As shown by our experiments in Appendix D.1, this is precisely what happens empirically when only optimizing the $r^{SS}$ term: resulting in a teacher that only learns to repeat the solution tokens themselves rather than providing general examples of reasoning steps that can be applied to new problems.
>
> Following the reviewer’s comments, we tried to make our motivation clearer by extending Section 3.3 as summarized above, explicitly explaining what would occur if we did not use the RLT format for $r^{KL}$:
>
> “[...] If we instead compared two distributions conditioned on the question alone, the KL would vanish and would fail to penalize the RLT steps that the teacher could not have generated without the solution in context.”
>
> > 5) Minor: Some sentences are too long to easily comprehend [W3.1]
>
> Following the reviewer’s feedback, we scanned through the text and broke down overly long sentences to improve readability. Two examples include:
>
> Lines 57-59 in our introduction, which we modified to:
>
> “We train RLTs with dense rewards using the student’s log **probabilities. Intuitively, our rewards measure both the student’s**” understanding of each problem’s ground-truth solution from the teacher’s explanations, and the interpretability of the logical leaps in the explanations themselves.”
>
> Lines 254-256 in Section 4, which we modified to:
>
> “We also find the effectiveness of the RLT traces stays consistent across different data **sizes. In contrast, the R1 traces from the Bespoke pipeline appear** significantly less effective when subsampled.”
>
> If the reviewer has any other specific example where text clarity could be improved that we might have missed, we hope they will not hesitate to let us know.
>
> > 6) Minor: In Figure 1, unclear whether it is the student size or teacher size [W3.2]
>
> Following the reviewer’s feedback, we added a second sentence to the caption in Figure 1 to explicitly refer to the teacher/student sizes in each subplot:
>
> “A 7B RLT provides better distillation and RL cold-starts [...] **We show this holds both when distilling 7B students of the same size (Left) and also 32B students, much larger than the RLT itself (Right).**”
>
> **references**
>
> [1] Liu, Aixin, et al. "Deepseek-v3 technical report."
>
> [2] Guo, Daya, et al. "Deepseek-r1: Incentivizing reasoning capability in llms via reinforcement learning."
>
> [3] Li, Dacheng, et al. "LLMs Can Easily Learn to Reason from Demonstrations Structure, not content, is what matters!."
>
> [4] Bespoke Labs. Bespoke-stratos: The unreasonable effectiveness of reasoning distillation.
>
> [5] Ning, Xuefei, et al. "Can LLMs learn by teaching for better reasoning? A preliminary study."

---

> > ### Comment · Reviewer_UdmA · 2025-08-04
> >
> > Thank you to the authors for the detailed and thoughtful rebuttal. The response addresses most of my concerns and questions. I understand that scaling up the model may exceed the available time and compute budget during the rebuttal period, and I consider the scaling-down experiment a reasonable alternative. I am maintaining my initial score that leans toward the positive side, and encourage the authors to include a scaling-up experiment in the final version to further strengthen the paper.

---

> > > ### Author Response · Authors · 2025-08-04
> > > **Response to Reviewer comment**
> > >
> > > Unfortunately, few industry/academic labs have the resources to run RL reasoning for very long context sizes and large models beyond 7 billion parameters. This is because this setting would need to shift most of the GPUs away from the VLLM autoregressive generation simply to keep the model/optimizer in GPU memory without OOM crashes (291.16 GB of VRAM at 16K ctx for 7B model vs 580.28 GB of VRAM for a 14B model).
> > >
> > > In concrete terms, running such an experiment with a 14B model on our original single-node setup (8 x H100 GPUs, i.e., 8 x 80GB = 640 GB of VRAM total) would take over a month, which is far beyond the resources we have available for this project. In contrast, SFT is very fast and cheap, as no autoregressive generation loop is required (e.g., we trained our 32B students in less than a day).
> > >
> > > However, we still believe our results with a 7B RLT are very significant as we consistently improved the distillation and cold-starting performance of state-of-the-art expensive pipelines using the 671B DeepSeek R1 model and other closed-source LMs [1, 2] across all examined tasks and student model sizes.
> > >
> > > We hope this additional clarification about the bottlenecks of our long-context reasoning setting will help better contextualize the challenges of training with RL beyond a 7 billion parameter scale.
> > >
> > > Regardless, we would like to thank the reviewer once again for their original review and for engaging in the discussion with their latest response. We believe their scaling suggestion and the new 3B RLT experiment will provide new solid evidence for our work, confirming that larger teachers indeed provide stronger performance. We hope they will not hesitate to let us know for any additional comments or questions.
> > >
> > > [1] Li, Dacheng, et al. "LLMs Can Easily Learn to Reason from Demonstrations Structure, not content, is what matters!."
> > >
> > > [2] Bespoke Labs. Bespoke-stratos: The unreasonable effectiveness of reasoning distillation.

---

### Official Review · Reviewer_Esvg · 2025-07-01

**Clarity:** 3
**Significance:** 3
**Originality:** 3
**Rating:** 4
**Confidence:** 4

**Summary:**

This paper addresses the current challenges in reinforcement learning for training LLMs on reasoning tasks, such as exploration difficulties and poor distillation performance, by proposing a novel Reinforcement-Learned Teachers (RLTs) framework.
Unlike traditional RL models that solve problems from scratch, RLTs are explicitly provided with problems and their corresponding answers during training, generating explanatory processes that are pedagogically valuable for student models. The framework is optimized based on feedback from the student models.

**Questions:**

1. There appears to be a disconnect between the paper's core contribution and its title. From my perspective, the work primarily addresses the challenges of cold starts and sparse rewards in reinforcement learning. Could you please clarify the connection between these aspects and the concept of "Test-Time Scaling" as highlighted in the title?

2. Could you provide a more detailed explanation as to why the proposed method is considered scalable? What specific properties or mechanisms enable its scalability?

**Ethical Concerns:**

["NO or VERY MINOR ethics concerns only"]

**Final Justification:**

I previously had some misunderstandings about the role mechanism of teachers during the testing phase. I appreciate the author's response, which has now clarified the issue. I will improve my rating

**Limitations:**

yes

**Quality:**

3

**Strengths And Weaknesses:**

Strengths
1. The proposed method effectively addresses the critical problem of sparse rewards in reinforcement learning. A significant advantage is its compatibility with any existing RL algorithm, making it a flexible and practical contribution to the field.



Weakness
1. The RLT teacher model requires the standard answer to each problem as input in order to generate explanations. This means the method cannot be directly applied to unknown problems or real-world reasoning tasks without provided answers. During the testing phase, the student model still relies on explanations from the teacher to produce answers, lacking independent problem-solving ability. Additionally, the teacher's explanations may already include the answer, effectively revealing the solution to the student model—naturally leading to better performance.
2. Although the authors designed a reward function based on student model's feedback to optimize the teacher model, the training process remains a one-way offline optimization. The student model remains fixed during the teacher's training phase and cannot dynamically adapt or learn based on the explanations generated by the teacher. This may lead to issues such as the reward function overfitting to the initial parameters of a specific student or the teacher's training objectives failing to align in real time with the student's learning curve.
3.  No  ablation study was conducted to evaluate the independent contributions and sensitivity of different reward terms(r^SS vs. r^KL)

---

> ### Author Rebuttal · Authors · 2025-07-30
>
> We would like to thank Reviewer Esvg for their feedback and the time they dedicated to our review. We worked to address each of their comments and hope they will not hesitate to let us know of any further suggestions or questions.
>
> **weaknesses**
>
> > 1) During the testing phase, the student model still relies on explanations from the teacher to produce answers, lacking independent problem-solving ability. Additionally, the teacher's explanations may already include the answer, effectively revealing the solution to the student model—naturally leading to better performance.
>
> We would like to clarify that the teacher’s explanations are only used during the training of our students. During testing, our student models do not rely in any way on the explanation produced by the teacher and can effectively solve problems from scratch independently. In fact, our procedure for distilling and testing our students is exactly the same as prior distillation approaches [1, 2]. The only difference is how we obtain the distillation dataset:
> - Prior approaches [1, 2] using expensive LLMs not only make use of the ground-truth solutions to verify the correctness of their reasoning traces but also rely on filtering and post-processing stages to improve the quality of their distillation data.
> - In contrast, we show RLTs are more effective without requiring any filtering and post-processing stages, using the raw outputs from orders-of-magnitude smaller 7B teacher LMs to generate the distillation data.
>
> Following the reviewer’s feedback, we expanded the text in Section 4.1 by including the above explicit comparison with the traditional RL distillation pipeline, which we hope will make it clearer to readers that our RLT framework actually has lower requirements and that our students do possess stronger independent problem-solving ability.
>
> > 2) [...] The student model remains fixed during the teacher's training [...] This may lead to issues such as the reward function overfitting [...] or failing to align in real time [...]
>
> Throughout our experiments in Section 4, we found that keeping the student model fixed was sufficient for our 7B teacher to outperform distillation/cold-starting with LLMs such as DeepSeek R1. While optimizing both models at the same time would be more costly in terms of computation, we do agree with the reviewer that concurrent optimization is a potential further direction to scale up our framework. In fact, we mention this exact idea when discussing future extensions in Section 6:
>
> “One example [of future directions] is training RLTs and their students in tandem, allowing the teacher's explanations to adapt to the student's learning dynamics live” (lines 361-363).
>
> Nonetheless, to still try to address the reviewer’s feedback, we collected and analyzed two new sets of more expensive experiments in Appendix D, taking a first step in this direction:
> 1. First, we considered conducting training in two phases, each optimizing both teacher and student for half the total number of optimization steps. In particular, we paused the teacher's RL optimization midway, distilled a student with this preliminary RLT’s explanations, and resumed our teacher optimization for the remaining steps with this updated student to compute the RLT rewards. The objective of this experiment is precisely to lessen the risk of potential overfitting to the student’s initial parameters, as suggested by the reviewer.
> 2. Second, we also considered using both the 7B and the 32B students for computing the RLT reward, averaging their probabilities. The objective of this experiment was instead to lessen the risk of potential overfitting to a single model.
>
> |Evaluated model|Teacher|AIME|MATH 500|GPQA|Overall|
> |-|-|-|-|-|-|
> |Qwen2.5-7B|N/A|10.0|74.2|33.3|39.2|
> |Bespoke-7B|DeepSeek R1|20.0|82.0|37.8|46.6|
> |RLT-7B|RLT teacher|23.3|82.8|42.4|49.5|
> |RLT-7B(2-stage teacher)|RLT teacher(2-stage training)|26.7|84.0|41.4|50.7|
> |RLT-7B(7B + 32B $r^{RLT}$)|RLT teacher(7B + 32B $r^{RLT}$)|23.3|83.8|41.9|49.7|
>
> As summarized in the Table above, both these extensions provide some moderate additional improvement to the RLT's distillation performance. However, we also note that these improvements came at a substantial additional cost, taking approximately 1.8x and 1.4x more GPU hours than our original RL training pipeline.
>
> > 3) No ablation study was conducted to evaluate the independent contributions and sensitivity of different reward terms.
>
> We would like to clarify that we did perform an in-depth ablation of our reward terms in Section D of our Appendix:
> 1. First, we examined ablating the think tokens KL reward term $r^{KL}$, quantifying whether the think tokens $t_{o_i}$ themselves are interpretable logical continuations from the student's perspective.
> 2. Second, we examined ablating the min/max reductions terms, which serve to ensure the rewards do not forego any individual token, as detailed in Section 3.3.
>
> We provided these results in Table 7 and summarize them again in the Table below for easier reference:
>
> |Evaluated model|AIME|MATH 500|GPQA|Overall|
> |-|-|-|-|-|
> |Qwen2.5-7B|10.0|74.2|33.3|39.2|
> |Bespoke-7B(R1 traces)|20.0|82.0|37.8|46.6|
> |Full RLT reward|23.3|82.8|42.4|49.5|
> |No thought tokens KL term $r^{KL}$|6.7|63.8|31.8|34.1|
> |No min/max reward reduction|23.3|79.0|40.0|47.4|
>
> Furthermore, beyond performance, in Figures 8-12, we also compared the lengths of the produced teacher explanations and provided several qualitative, concrete examples showing the effects of each reward term. As detailed in Appendix D.1, ablating $r^{KL}$ leads to a teacher that only learns to repeat the solution tokens themselves in its explanation to exploit the repetition tendency of pretrained student LMs, with the distillation performance and average length of its output reasoning traces dramatically dropping. Furthermore, ablating the min/max reward reduction terms makes the rewards effectively biased by the length of the reasoning trace. While this second ablation only minimally influences performance, it makes the number of think tokens in the teacher's reasoning traces almost double, unnecessarily increasing the downstream student distillation cost.
>
> Following the reviewer’s feedback, we extended lines 198-199 at the end of Section 3.3 to reference more explicitly the presence of this set of important results that validate our reward design, and avoid readers potentially missing them:
>
> “For further discussion, we refer to Appendix D, where we empirically analyze and validate each design choice in our reward function in terms of final performance (Table 7) and concrete qualitative differences of the resulting explanations (Figures 8-11).”
>
> **questions**
>
> > 1) Could you please clarify the connection between these aspects and the concept of "Test-Time Scaling" as highlighted in the title?
>
> By “test-time scaling,” we are referring to the RLT’s ability to produce explanations for training students and providing them the ability to use additional compute to generate lengthy reasoning traces and improve performance. We note that this same nomenclature is used in the  abstracts, introductions, and titles of many prior important papers in the LM post-training field, such as DeepSeek-R1 [3], THUDM’s T1 [4], and "s1: Simple test-time scaling." [5].
>
> Following the reviewer’s comment, we tried making the connection more evident for our readers by modifying lines 26-29 and adding direct citations to the above example works:
>
> “With the rise of RL for open-ended reasoning (RL reasoning), inducing a new form of language model (LM) test-time scaling [3, 4, 5]...”
>
>  > 2) Could you provide a more detailed explanation as to why the proposed method is considered scalable? What specific properties or mechanisms enable its scalability?
>
> By simplifying the RL task and directly aligning the RL objective with downstream student learning, we show how a 7B RLT provides better distillation and cold-starting data than the 671B DeepSeek R1 model and other closed-source LMs. These results provide concrete evidence that the RLT framework has the potential to become a more scalable framework by greatly reducing the cost of training advanced models. Instead of relying on large models at every stage, our framework allows training small, specialized teachers that can be used to teach much larger models efficiently. This flips the traditional scaling paradigm: the heaviest work (RL training) can now be handled by compact, more affordable models.
>
> To better put the scalability and efficiency of our framework into perspective, we added a section comparing the GPU cost of training and collecting reasoning traces for distillation with our RLT model compared with an estimated lower-bound cost of RL training and data collection using DeepSeek R1 (see our first response to reviewer UdmA for details):
>
> |Model/GPU-hours|Training(GPU model)|Data generation(GPU model)|
> |-|-|-|
> |DeepSeek R1|>688000(H800)|>1067(H100)|
> |7B RLT teacher|280.4(H100)|6.7(H100)|
>
> We hope that adding this new direct comparison and discussion to our paper will help better contextualize the scalability and potential of our new RLT framework, allowing for cheaply distilling large reasoning models while performing expensive RL training with much smaller and efficient ones to counteract the exponentially growing training costs from the scaling laws of LLMs [6].
>
> **references**
>
> [1] Li, Dacheng, et al. "LLMs Can Easily Learn to Reason from Demonstrations Structure, not content, is what matters!."
>
> [2] Bespoke Labs. Bespoke-stratos: The unreasonable effectiveness of reasoning distillation.
>
> [3] Guo, Daya, et al. "Deepseek-r1: Incentivizing reasoning capability in llms via reinforcement learning."
>
> [4] Hou, Zhenyu, et al. "Advancing language model reasoning through reinforcement learning and inference scaling."
>
> [5] Muennighoff, Niklas, et al. "s1: Simple test-time scaling."
>
> [6] Hoffmann, Jordan, et al. "Training compute-optimal large language models."

---

### Official Review · Reviewer_YrSD · 2025-07-04

**Clarity:** 3
**Significance:** 4
**Originality:** 4
**Rating:** 5
**Confidence:** 4

**Summary:**

The authors introduce the RLT paradigm where the teacher's task is to simply provide explanations for a student given a question-answer pair, rather than prior approaches where the teacher takes in only the question to give out explanations and the solutions. This approach is intuitive and cleverly circumvents the exploration problem from correctness-based sparse rewards that are typically used. The authors demonstrate multiple use cases and advantages of the RLT paradigm: efficiently training larger students with a smaller teacher model, showing out-of-domain transfer for the teaching task, applying RLT to cold-start future RL iterations. The RLT paradigm is instantiated in the GRPO setting where the sparse based reward is a replaced with a dense-reward modelling two aspects: student understanding of the the solution given the question, and agreement of the teacher logical explanation under the student model and teacher model.

**Questions:**

1. The authors state that the paradigm of solely correctness-based rewards does not allow the model to explore beyond its initial latent abilities (line 38). With the RLT's dense reward paradigm, can there be exploration beyond the initial LM's latent abilities? To me, this answer still seems to be "no" -- but since the teacher does not have to solve the task, there are no sparse-reward problems. If this is indeed the case, the authors must clarify this in the paper.

**Ethical Concerns:**

["NO or VERY MINOR ethics concerns only"]

**Final Justification:**

I have read the rebuttal and my questions/concerns have been sufficiently addressed. Since my original score was already high and recommending acceptance, I will keep my score (5).

**Limitations:**

Yes

**Quality:**

4

**Strengths And Weaknesses:**

Strengths:
1. The RLT paradigm cleverly avoids the exploration problem by modeling the teaching task with a dense-reward objective, rather than the usual correctness based rewards that are sparse.
2. Intuitively, the task of providing an explanation given a question-answer pair seems much simpler and makes a lot of sense. This is reflected in a smaller teacher able to effectively train a much larger student.
3. Out-of-domain zero-shot RLT outperforming direct RL on the countdown task is a strong and interesting result.
4. The authors also provide the code and many details of their approach in the paper, so the work seems to be reproducible.

Weaknesses:
1. The authors state that the RLT framework can be used with any RL algorithm, but the experiments performed solely use GRPO. Although I find this believable, the paper could benefit from an experiment to demonstrate and verify this claim.
2. The authors do not discuss what happens in cases where there is a large mismatch in the abilities of the teacher and student LM. For example, a small teacher LM might output explanations that are not sound (while still maintaining a high logprob under its own distribution). This can be exacerbated in scenarios where the student LM is much larger (or a same-size student that is trained to solve a domain extremely well). Will the RLT paradigm then resort to reward-hacking (i.e., will the student over-trust the teacher's explanations regardless of its own initial abilities)? An ablation on "small teacher LM", with increasingly more capable student LMs (beyond 32B) would greatly improve the analysis of the paper.
3. In (i) of section 3.3, the authors state: _This first reward term is computed with the student’s log probabilities over the solution tokens, reduced with both average and minimum operations_ and then provide equation 3 which seems to imply that the reward is the average log prob + $\alpha$ min logprob. From the text, I interpreted that the logprobs are normalized (divided by) the avg and min logprobs. This hinders clarity. Similar comments apply to also (ii) in section 3.3, and equation 4. The authors can consider modifying text to improve clarity.

Overall, I remain positive about the paper and believe it provides many new insights.

---

> ### Author Rebuttal · Authors · 2025-07-30
>
> We would like to thank Reviewer YrSD for their feedback and the time they dedicated to our review. We worked to address each of their comments and hope they will not hesitate to let us know of any further suggestions or questions.
>
> **weaknesses**
>
> > 1) The authors state that the RLT framework can be used with any RL algorithm, but the experiments performed solely use GRPO. Although I find this believable, the paper could benefit from an experiment to demonstrate and verify this claim.
>
> Following the reviewer’s suggestion, we extended our implementation and added a new set of experiments training our 7B RLT model using the RLOO [1] algorithm instead of GRPO [2]. We added these new results and analysis to Appendix C, which we summarize below:
>
> |Evaluated model|Teacher model|AIME 2024|MATH 500|GPQA Diamond|Overall|
> |-|-|-|-|-|-|
> |Qwen2.5-7B-Instruct|N/A|10.0|74.2|33.3|39.2|
> |Bespoke-7B|DeepSeek R1|20.0|82.0|37.8|46.6|
> |RLT-7B|RLT teacher (GRPO)|23.3|82.8|42.4|49.5|
> |RLT-7B (RLOO teacher)|RLT teacher (RLOO)|20.0|83.6|42.9|48.8|
>
> In line with recent empirical findings about the similar empirical effectiveness of the different reasoning RL algorithms [3], the performance of our RLT trained with RLOO appears very close to our previous results obtained using GRPO. We believe the small gap is mostly due to the fact that we did not re-adjust any of the main hyperparameters from our GRPO implementation (e.g., batch size, number of generations per question). We hope these new results will provide concrete evidence confirming that the RLT paradigm's performance is not tied to any specific RL algorithm.
>
> > 2) The authors do not discuss what happens in cases where there is a large mismatch in the abilities of the teacher and student LM [...] An ablation on "small teacher LM", with increasingly more capable student LMs (beyond 32B) would greatly improve the analysis of the paper.
>
> While our current experiments do show that even with a significantly smaller 7B teacher, our framework remains effective for teaching a much larger student, we understand the reviewer’s interest in examining the effect of pushing the gap between teacher and student to the extreme. However, while we ran all our experiments on a single H100 node (8 GPUs), we note that training models beyond 32B would require multi-node training and be computationally too expensive for this project’s resources.
>
> Thus, to still try to address the reviewer’s suggestion, we instead focused on further decreasing the size of the RLT teacher down to 3B parameters. We added these new results to Appendix D of our current revision, which we summarize in the Table below:
>
> |Evaluated model|Teacher model|AIME 2024|MATH 500|GPQA Diamond|Overall|
> |-|-|-|-|-|-|
> |Qwen2.5-7B-Instruct|N/A|10.0|74.2|33.3|39.2|
> |RLT-7B|RLT teacher(7B)|23.3|82.8|42.4|49.5|
> |RLT-7B(3B teacher)|RLT teacher(3B)|20.0|80.6|38.9|46.5|
> |Qwen2.5-32B-Instruct|N/A|26.7|84.0|49.0|53.2|
> |RLT-32B|RLT teacher(7B)|66.7|93.4|59.6|73.2|
> |RLT-32B(3B teacher)|RLT teacher(3B)|46.7|91.4|53.5|63.9|
>
> As shown by our new set of results, we expectedly find that larger teachers yield better explanations and downstream results. However, we find that most of the performance gap between our 3B and 7B teachers only occurs when distilling the 32B student LM, where the mismatch between teacher and student capabilities tends to the extreme. However, even in this case where the student is over 10x larger than the teacher, our 3B RLT is still able to provide considerable improvements to the initial Qwen 32B student.
>
> The reason that we did not observe performance degradation appears to be due to the role of the $r^{KL}$ reward term plays during the optimization. Even in cases where the initial 3B teacher is unable to provide logical explanations to a specific question and answer, optimizing $r^{KL}$ will naturally guide the teacher's output distribution (with both question and answer in context) toward converging to what was most likely from the student's distribution (with only the question in context). We added in Appendix D an analysis of this phenomenon, which we validated by inspecting at different training stages the 3B teacher's output explanations for challenging questions with low values of $r^{SS}$.
>
> We believe these results further validate how the RLT framework enables even small, inexpensive models to be effective teachers, but also confirm that larger teacher models are indeed able to provide better explanations, highlighting the potential future implications of scaling with more training compute.
>
> We are open to collecting additional results for future revisions of our work by pushing the gap between teacher and student even further by training and evaluating a 0.5B teacher model, in case the reviewer finds this further necessary.
>
> > 3) In (i) of section 3.3, [...] I interpreted that the logprobs are normalized (divided by) the avg and min logprobs. [...] Similar comments apply to also (ii) in section 3.3.
>
> Following the reviewer’s feedback, we went through the paper and extended the text in several cases where we previously used concise language (for saving space) to improve clarity. For example, in the specific instances flagged by the reviewer, we rewrote lines 175-176 and 178-181 to:
>
> “This first reward term is computed with the student’s log probabilities over the solution tokens. We reduce this vector to a scalar by applying both average and minimum operations over the different per-token values:” (ex lines 175-176)
>
> “This second reward term is computed with the KL divergence over the same think tokens [...].
> Similarly to $r^{SS}$, we reduce this vector to a scalar by applying both average and maximum operations over the different per-token values:” (ex lines 178-181)
>
> **questions**
>
> > 1) The authors state that the paradigm of solely correctness-based rewards does not allow the model to explore beyond its initial latent abilities (line 38). With the RLT's dense reward paradigm, can there be exploration beyond the initial LM's latent abilities? To me, this answer still seems to be "no" -- but since the teacher does not have to solve the task, there are no sparse-reward problems [...]
>
> Following the reviewer's question, we realized that the term "exploration" could have different interpretations and added an explicit explanation in Section 1 and Appendix E for our statement that “correctness-based rewards do not allow the model to explore beyond its initial latent abilities,” which we summarize below:
>
> In traditional RL for robotics, neural network policies can learn to solve tasks from random initializations thanks to being guided by dense reward functions. Without this guidance, they would be faced with an infeasible exploration challenge made exponentially more difficult by the task horizon [4]. This is because dense rewards allow the policy gradient optimization to rank the relative progress obtained across its initial suboptimal actions, allowing the policy to bootstrap from partial solutions and extrapolate far beyond its initial capabilities.
>
> However, in the context of LM reasoning, as correctness-based rewards are inherently sparse, this extrapolation is not possible. In particular, if an LM is too weak to provide the optimal solution for a task, all its rewards (and, thus, the policy gradient) will be zero. In contrast, this limitation would not apply to our RLT, which can make use of a dense reward function and obtain a learning signal to iteratively improve even when all sampled initial explanations are suboptimal, as with the traditional RL framework.
>
> We hope this new discussion will clarify what we mean by “extrapolation beyond the LM’s initial latent abilities” (lines 37-38) and hope the reviewer will not hesitate to let us know if they believe anything else requires additional contextualization.
>
> **references**
>
> [1] Ahmadian, Arash, et al. "Back to basics: Revisiting reinforce style optimization for learning from human feedback in llms."
>
> [2] Shao, Zhihong, et al. "Deepseekmath: Pushing the limits of mathematical reasoning in open language models."
>
> [3] Hu, Jian, et al. "Reinforce++: An efficient rlhf algorithm with robustness to both prompt and reward models."
>
> [4] Devidze, Rati, et al. "Explicable reward design for reinforcement learning agents."

---

### Comment · Area_Chair_5wFe · 2025-08-04
**Gentle Reminder: Please Reply to Authors’ Responses (Only if Not Yet Done)**

Dear Reviewers,

As the discussion deadline approaches, may we kindly ask you to review the authors’ responses and post a constructive reply—unless you have already done so, in which case please kindly disregard this gentle reminder.

Your thoughtful engagement is deeply appreciated and essential to a fair and timely process. With sincere thanks for your continued dedication.

Area Chair

---

### Note · Authors · 2025-08-13

We appreciated the positive reception of the RLT framework introduced in our submission. In particular, we thank the reviewers for recognizing its conceptual novelty and the significance of our results, which showed that our 7B RLT teachers can distill and cold-start larger 32B LM students, generalize zero-shot to new domains, and even outperform prior state-of-the-art pipelines using orders-of-magnitude larger models such as GPT4, Gemini, and DeepSeek R1 [1, 2]. Nonetheless, we were pleased to receive several constructive suggestions, based on which we took concrete action during the rebuttal phase:

- We added experiments training RLTs with RLOO and using a 3B base model to provide further analysis on generality and scalability (YrSD, UdmA)
- We added experiments using multiple students for the RLT rewards, updating the student in the middle of training, and changing the amount of distillation data from NuminaMath (Esvg, UdmA, YMHP)
- We made adjustments to the text: addressing specific examples of long sentences, improving the clarity of our reward description, better defining our notions of "exploration" and "test-time scaling", and adding references to additional related works [3, 4, 5] (YrSD, Esvg, UdmA, YMHP)
- We added clearer pointers from the main text to our reward ablation analysis in Appendix D, and included a new direct comparison of the GPU hours for training and distillation using our 7B RLTs and our much more expensive baseline models (Esvg, YMHP, UdmA)

Following the discussion phase, we are grateful to reviewers YrSD, UDMA, and YMHP for posting final comments and confirming that our rebuttal successfully addressed their concerns and questions. Thanks to their concluding remarks:

- We added additional clarification to our conclusion to contextualize the bottlenecks of long-context reasoning and the prohibitive costs of training with RL beyond a 7B parameter scale (UdmA)
- We moved part of our reward ablation results in the main text for potential future revisions without strict page limits (YMHP)

We are also grateful to reviewer Esvg, who first submitted the mandatory acknowledgement and soon raised their confidence score after our rebuttal. We hope to have correctly interpreted this non-textual communication as a sign that we successfully addressed their feedback as well.

[1] LLMs Can Easily Learn to Reason

[2] s1: Simple test-time scaling

[4] Montessori-instruct

[5] Dataenvgym

[6] Can LLMs learn by teaching for better reasoning?

---

### Decision · Program_Chairs · 2025-09-17

**Decision:**

Accept (poster)

**Comment:**

Summary of the paper: The paper proposes Reinforcement-Learned Teachers (RLTs), a new training paradigm in which a teacher language model is not asked to solve a problem from scratch, but only to generate detailed, pedagogical explanations (CoT) for a given question–answer pair.  Under this framing the teacher’s reward is dense: it measures (i) how much the explanation raises the student’s log-probability of the correct answer and (ii) how probable the explanation itself is under the student’s current policy.  Trained with these two lightweight losses, a 7 B-parameter RLT produces raw reasoning traces that (1) outperform distilled traces from much larger teacher models (DeepSeek-R1, Gemini) when used to train students of any size, (2) serve as a superior cold-start for subsequent sparse-reward RL, and (3) generalise zero-shot to new reasoning domains (Countdown, Game-of-24, etc.).  The authors also show that a small teacher can efficiently supervise a far larger student, and that the method can be bolted onto any existing RL algorithm.


Strength of the paper:
1. Conceptual novelty and clarity: RLTs reframe the teacher’s job from “solve” to “explain given the solution” which is simple, intuitive, and pedagogically grounded. RLTs elegantly sidestep the well-known exploration difficulty of sparse rewards in reasoning tasks.
2. Empirical strength and breadth: RLTs demonstrate that a smaller teacher can teach a larger student, overturning the usual distillation assumption. RLTs also provide multiple use-cases: standard distillation, cold-starting RL, and zero-shot transfer, all supported by strong quantitative results. Out-of-domain transfer (Countdown task) shows generalization beyond the training distribution.
3. Practical impact and reproducibility: RLTs are compatible with any existing RL algorithm, making them an immediate plug-in improvement. Authors release code and detailed experimental protocols, satisfying reproducibility standards.

Weaknesses of the paper: After reading the rebuttal, I think the authors' response addresses most of the reviewers' concerns and questions. No major weaknesses are identified for this paper. Please incorporate the content during rebuttal into the camera-ready version of the paper.

Reasons for the decision:
1. I think this paper offers a fresh perspective on how to unlock the reasoning power of LLMs. RLT attacks a central pain-point in LRM training—sparse rewards and poor exploration.
2. The proposed losses are simple, theoretically grounded, and empirically validated across diverse settings. Specifically, a 7B RLT beats traces from much larger models, and zero-shot transfer to new tasks is demonstrated.
3. All four reviewers are positive, citing strong novelty, clarity, and experimental evidence, giving a clear consensus to accept.